# Plk1 overexpression induces chromosomal instability and suppresses tumor development

Guillermo de Cárcer [1], Sharavan Vishaan Venkateswaran [2,3], Lorena Salgueiro[2], Aicha El Bakkali[1], Kalman Somogyi[2], Konstantina Rowald[2], Pablo Montañés [1], Manuel Sanclemente[1], Beatriz Escobar[1], Alba de Martino [4], Nicholas McGranahan[5], Marcos Malumbres [1] & Rocío Sotillo [2,6]

Polo-like kinase 1 (Plk1) is overexpressed in a wide spectrum of human tumors, being frequently considered as an oncogene and an attractive cancer target. However, its contribution to tumor development is unclear. Using a new inducible knock-in mouse model we report here that Plk1 overexpression results in abnormal chromosome segregation and cytokinesis, generating polyploid cells with reduced proliferative potential. Mechanistically, these cytokinesis defects correlate with defective loading of Cep55 and ESCRT complexes to the abscission bridge, in a Plk1 kinase-dependent manner. In vivo, Plk1 overexpression prevents the development of Kras-induced and Her2-induced mammary gland tumors, in the presence of increased rates of chromosome instability. In patients, Plk1 overexpression correlates with improved survival in specific breast cancer subtypes. Therefore, despite the therapeutic benefits of inhibiting Plk1 due to its essential role in tumor cell cycles, Plk1 overexpression has tumor-suppressive properties by perturbing mitotic progression and cytokinesis.

[1] Cell Division and Cancer Group, Spanish National Cancer Research Centre (CNIO), Melchor Fernández Almagro 3, E-28029 Madrid, Spain. [2] Division of Molecular Thoracic Oncology, German Cancer Research Center (DKFZ), Im Neuenheimer Feld 280, 69120 Heidelberg, Germany. [3] Faculty of Biosciences, Heidelberg University, 69117 Heidelberg, Germany. [4] Histopathology Unit, Spanish National Cancer Research Centre (CNIO), 28029 Madrid, Spain. [5] Cancer Research UK Lung Cancer Center of Excellence, University College London Cancer Institute, Paul O'Gorman Building, 72 Huntley Street, London WC1E 6BT, UK. [6] Translational Lung Research Center Heidelberg (TRLC), German Center for Lung Research (DZL), Heidelberg, Germany. These authors contributed equally: Guillermo de Cárcer, Sharavan Vishaan Venkateswaran. Correspondence and requests for materials should be addressed to G.d.C (email: gcarcer@cnio.es) or to M.M. (email: malumbres@cnio.es) or to R.S. (email: r.sotillo@dkfz-heidelberg.de)

Chromosomal instability (CIN) is a frequent feature both in solid and hematopoietic human tumors[1,2]. Although its causal role during tumor development is still under careful experimental scrutiny, it is now clear that CIN provides specific clones with a variety of chromosomal combinations that may favor either tumor growth or resistance to antitumor therapies[3–5]. Multiple oncogenic alterations may induce CIN, although the copy number aberrations that ultimately arise do so as a consequence of defects in the cellular machinery that regulates chromosome segregation and protects from unequal chromosome inheritance during mitosis[1,2]. Whether alteration in the levels of the encoded proteins is a cause or consequence of CIN is not clear, although experimental overexpression of several components of the CIN signature such as Mad2[6], cyclin B1 and cyclin B2[7], as well as Aurora B[8] induces CIN and spontaneous tumor formation in mouse models[9].

Plk1 is the most studied member of a conserved family of protein kinases (Plk1–5) involved in cell division as well as specific functions in postmitotic cells such as neurons[10] or smooth muscle cells[11]. Plk1 was originally identified in *Drosophila* as a protein involved in spindle formation and further studies have suggested critical functions for this kinase in centrosome biology, spindle dynamics, chromosome segregation, and cytokinesis[12,13]. Genetic ablation of *Plk1* or its chemical inhibition results in defective chromosome segregation commonly accompanied by cell cycle arrest or cell death in a variety of model organisms[13,14]. Plk1 induction has been proposed to play a role at early stages during the progression of certain carcinomas and its overexpression inversely correlates with the survival rate of patients with non-small cell lung, head and neck, and esophageal cancer, among others[15–17]. Plk1 inhibition with specific small molecule inhibitors is currently considered as an attractive therapeutic strategy against specific tumor types such as leukemia and non-small cell lung cancer[18–20].

From the previous studies, Plk1 has been frequently considered as a classical oncogene. However, the cellular effects of Plk1 overexpression in malignant transformation and their implications in tumor development have not been analyzed. In this study, we found that Plk1 overexpression functions as a tumor suppressor both in vitro and in vivo. Elevated levels of Plk1 delay mammary gland tumor formation driven by classical oncogenes such as Kras[G12D] or Her2. At the cellular level, these effects are accompanied by multiple aberrations during mitosis, as well as impaired loading of ESCRT complexes during cytokinesis because of increased Plk1 kinase activity. Importantly, increased levels of Plk1 in breast cancer patients is associated with better prognosis.

## Results

**A new mouse model for inducible Plk1 overexpression.** To investigate the consequences of Plk1 overexpression we first generated KH2 mouse embryonic stem (ES) cells[21] in which a FLAG-tagged human Plk1 cDNA was introduced downstream of the ColA1 gene (Fig. 1a). In this construct, the FLAG-Plk1 cDNA is expressed under the tetracycline-inducible operator (tetO) sequences and it is therefore induced after the activation of the reverse tetracycline transactivator (rtTA; expressed in the *Rosa26 locus*) with the tetracycline derivative doxycycline (Dox; Fig. 1a). Treatment of these ES cells with Dox resulted in rapid induction of FLAG-Plk1 (Fig. 1b), which was detected in the spindle poles and the spindle during mitosis suggesting a proper localization of the encoded FLAG-Plk1 protein (Fig. 1c). We then generated heterozygous (referred to as ColA1(+/Plk1) or (+/Plk1) in brief) or homozygous (Plk1/Plk1) knockin mice after microinjection of these ES cells into developing morulas. These knockin mice also expressed the *Rosa26-rtTA* (referred to as *rtTA*) transactivator

either in heterozygocity (+/rtTA) or homozygocity (rtTA/rtTA) (Fig. 1a), resulting in an efficient induction of FLAG-Plk1 expression in different tissues after administration of Dox (Fig. 1d and Supplementary Fig. 1a).

We first induced FLAG-Plk1 expression in vivo using heterozygous or homozygous knockin mice in the presence of two copies of the transactivator [(+/Plk1); (rtTA/rtTA) or (Plk1/Plk1); (rtTA/rtTA), respectively]. Unexpectedly, most of these animals died (Supplementary Fig. 1b) as a consequence of a rapid loss of weight (Supplementary Fig. 1c) during the first weeks on the Dox diet and with reduced counts of red blood cells, white blood cells, and lymphocytes (Supplementary Fig. 1d). Double heterozygous mutants [(+/Plk1);(+/rtTA)] treated with Dox since birth displayed a slightly reduced tumor-free survival (Fig. 1e), accompanied by a slight but non-significant increase in some tumors such as lymphomas and sarcomas (Supplementary Fig. 1e, f). Overexpression of Plk1 in Dox-treated (+/Plk1);(+/rtTA) mice was accompanied by alteration in the nuclear size and loss of architecture in some tissues such as bronchial epithelia, pancreas, or liver (Fig. 1f).

**Plk1 overexpression impairs proliferation and transformation in vitro.** To analyze the cellular consequences of Plk1 overexpression during cell division, we next used (+/Plk1);(+/rtTA) or (Plk1/Plk1);(+/rtTA) mouse embryonic fibroblasts (MEFs) generated from these mutant mice. The transgenic Plk1 was properly located in these Dox-treated MEFs, showing cytoplasmic and nuclear localization in interphase cells with preferential localization to the centrosomes, as well as decorating the spindle poles, kinetochores, and the cytokinesis midbody in mitotic cells (Supplementary Fig. 2a, b). Induction of Plk1 at intermediate or high levels in MEFs (Fig. 2a) resulted in defective cell proliferation (Fig. 2b–d), accompanied by the formation of higher number of polyploid cells that accumulated with time (Fig. 2e, f) without increasing apoptosis (Supplementary Fig. 2c). Plk1 overexpression also induced upregulation of the tumor suppressor p53 as well as its target p21[Cip1] (Supplementary Fig. 2d) and led to the generation of senescent cells characterized by the expression of senescence-associated β-galactosidase and flat morphology (Supplementary Fig. 2e).

We next asked whether Plk1 overexpression could cooperate with oncogenic transformation by HrasV12 in immortal MEFs. As depicted in Fig. 2g, Plk1 overexpression dramatically reduced the number of Ras-transformed foci, and these transformed cells were unable to grow in soft agar (Fig. 2h). Additionally, HrasV12 and E1A transformed primary MEFs also grow slower after Dox administration compared to uninduced clones (Supplementary Fig. 2f, g) and Plk1 expression continued to generate binucleated cells (Supplementary Fig. 2h), despite being transformed. Finally, to provide further evidence that Plk1 overexpression facilitates polyploidization in non-transformed human cells, MCF10A-rtTA cells were infected with a Dox inducible Plk1 vector. As shown in Supplementary Fig. 2i, Plk1 overexpression resulted in a significant increase in binucleated cells after 48 h on Dox. All together, these results suggest an antiproliferative effect of Plk1 overexpression in these assays.

We also checked whether the negative effects of Plk1 overexpression were uniquely present in p53-proficient cells. We generated MEFs derived from (+/Plk1); (rtTA/rtTA); p53 (−/−) and (Plk1/Plk1); (rtTA/rtTA); p53(−/−) mice and tested the effect of Dox-mediated induction of Plk1. Plk1 overexpression in p53-null cells also resulted in deficient cell proliferation (Supplementary Fig. 3a) in the presence of highly polyploid cells (Supplementary Fig. 3b). In addition, lack of p53 did not rescue the defects in cell transformation induced by Plk1 overexpression

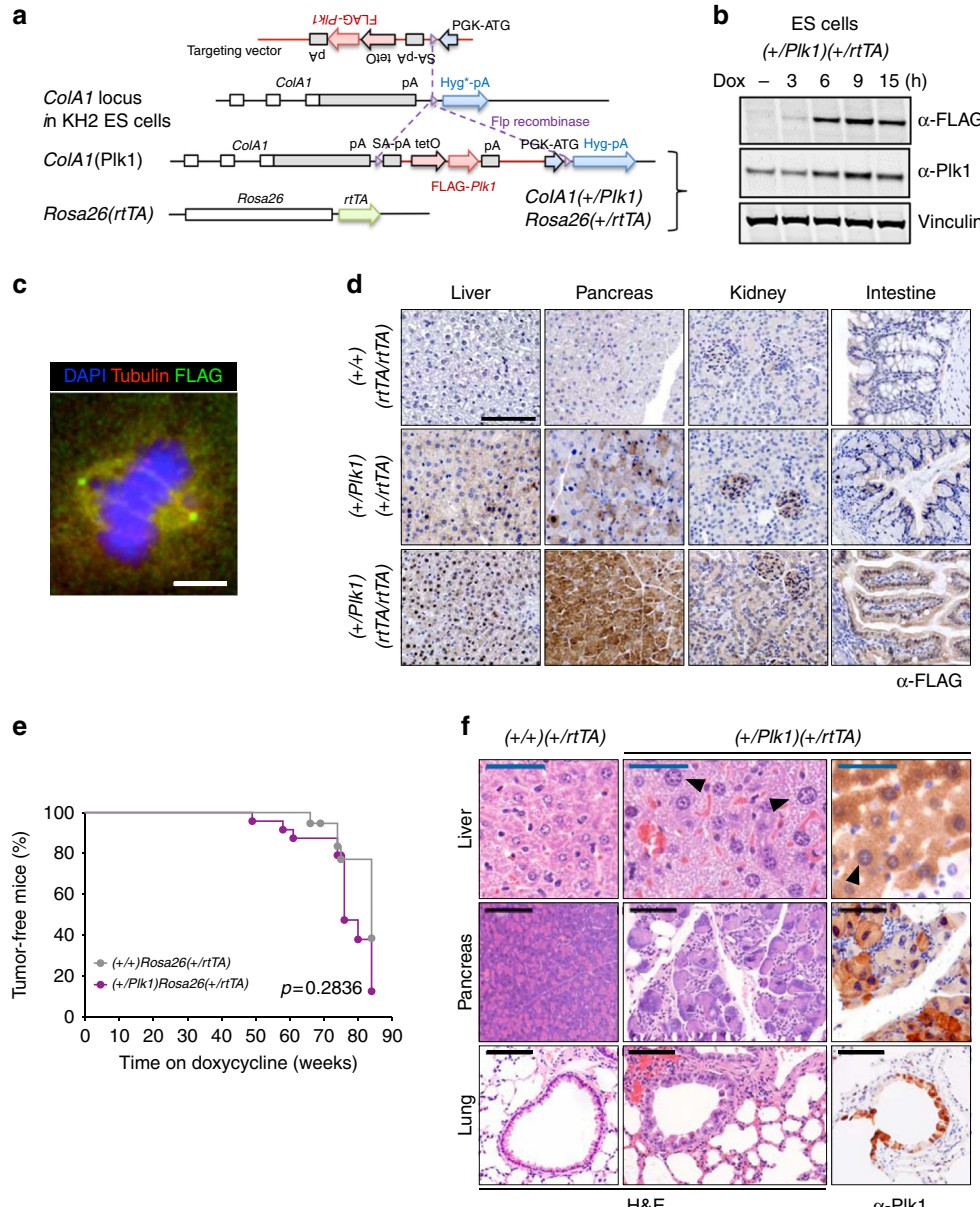

**Fig. 1** Generation of Plk1-inducible mice. **a** Schematic representation of the alleles used in this work. A cassette containing the human FLAG-Plk1 cDNA downstream of the tetO sequences is inserted in the endogenous *ColA1* locus after homologous recombination in KH2 ES cells. This allele [*ColA1*(Plk1) or (Plk1) in short] is later combined with the Rosa26-rtTA allele expressing the tetracycline transactivator. **b** (+/Plk1);(+/rtTA) ES cells were treated with Dox and FLAG and Plk1 signal was detected using specific antibodies at the indicated time points. Vinculin was used as a loading control. **c** Immunofluorescence of (+/Plk1);(+/rtTA) ES cells treated with Dox for 12 h. FLAG (green) is concentrated at the spindle poles with some signal in the spindle microtubules and additional diffuse signal as expected for Plk1. α-tubulin is in red and DAPI in blue. Scale bar 5 μm. **d** Immunodetection of Flag-Plk1 in the indicated tissues from (+/+);(rtTA/rtTA); (+/Plk1)(+/rtTA) and (+/Plk1)(rtTA/rtTA) mice treated with Dox for 8 weeks. Scale bar 100 μm. **e** Tumor-free survival of (+/+); (+/rtTA) and (+/Plk1)(+/rtTA) mice fed with Dox since birth during 85 weeks. (+/+)(+/rtTA), 19 mice; (+/Plk1)(+/rtTA), 24 mice. *p* = 0.2836; Log-rank (Mantel–Cox) test. **f** Sections of Dox-treated (+/+);(+/rtTA) and (+/Plk1)(+/rtTA) mice after staining with hematoxylin and eosin (H&E) or immunodetection of Plk1 (right panel). Cells with abnormally large nuclei are indicated with arrows. Black scale bar: 100 μm, blue scale bar: 50 μm

in the presence of the HrasV12 oncogene (Supplementary Fig. 3c), suggesting that the anti-proliferative defects induced by Plk1 overexpression do not require an active p53-mediated response.

**Plk1 overexpression disrupts proper chromosome segregation and cytokinesis.** Since Plk1 has been involved both in DNA replication as well as in mitosis[22,23], we first tested whether Plk1 induction had any obvious effect in DNA replication. Treatment with Dox did not alter the number of cells entering S-phase at early time points in the first cell cycle after stimulation of cell cycle entry with serum (Supplementary Fig. 4a). In addition, we did not observe a significant increase of DNA damage foci (as detected with antibodies against phosphorylated (γ)-H2AX or 53BP1) after Plk1 overexpression, suggesting no major defects in DNA replication (Supplementary Fig. 4b, c).

We next followed progression throughout mitosis using time-lapse microscopy in cells co-expressing GFP-tagged histone H2B. Overexpression of Plk1 lead to a variety of mitotic defects such as monopolar and multipolar spindles in prometaphase, as well as

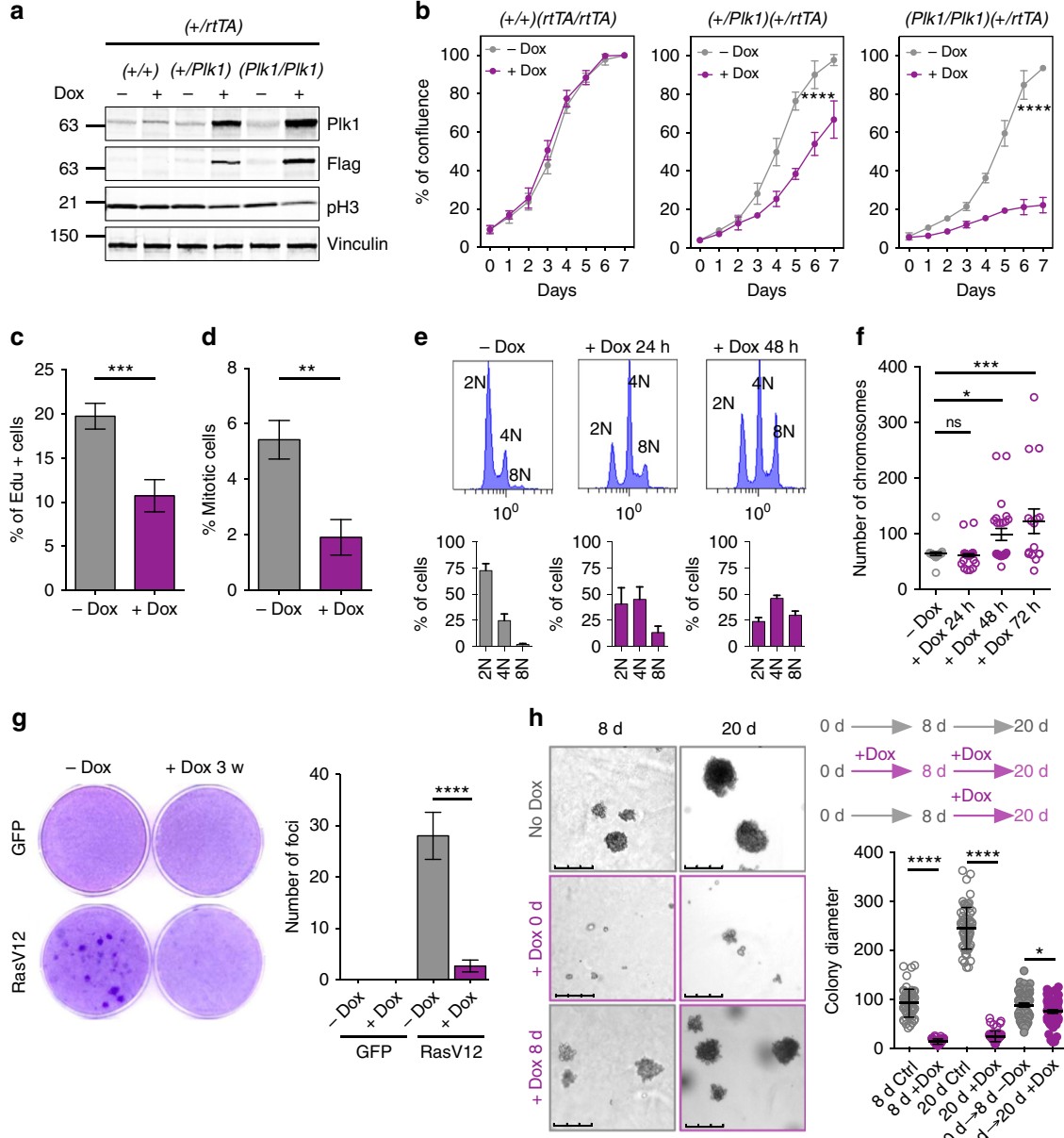

**Fig. 2** Plk1 overexpression results in proliferative defects in cultured MEFs. **a** Expression of Plk1 in MEFs with the indicated genotypes, after 24 h treatment with doxycycline. Cell lysates were immunoblotted with anti-Plk1, anti-Flag, anti-pH3, and anti-vinculin as a loading control. **b** Quantification of confluence in cultures with the indicated genotypes in the absence (−Dox) or presence (+Dox) of doxycycline for 7 days. ****$p < 0.0001$; two-way ANOVA. **c** Percentage of EdU-positive cells in (+/Plk1);(rtTA/rtTA) cultures untreated or treated with Dox for 3 days. Cells were exposed to EdU for 1 h before the analysis. ***$p < 0.001$; Student's $t$-test. **d** Percentage of mitotic cells in (+/Plk1);(rtTA/rtTA) cultures untreated or treated with Dox for 2 days as detected by phospho-Ser10 Histone H3 immunofluorescence. **$p < 0.05$; one-way ANOVA. **e** DNA content in (+/Plk1);(rtTA/rtTA) cells untreated or treated with Dox for 1 or 2 days. The percentage of 2N, 4N, or 8N cells is indicated in the histograms. **f** Metaphase spreads of (+/Plk1);(rtTA/rtTA) cells untreated or treated with Dox for the indicated time. The number of chromosomes per cell is shown in the plot ($n = 25$ (−Dox), $n = 41$ (24 h), $n = 25$ (48 h), $n = 16$ (72 h) cells per condition). ns not significant; *$p < 0.05$, ***$p < 0.001$; one-way ANOVA. **g** Focus formation assays of (+/Plk1);(rtTA/rtTA) MEFs transfected with oncogenic HrasV12 or control GFP-expressing vectors in the absence (−Dox) or presence (+Dox) of doxycycline. The number of foci is indicated in the histogram. ****$p < 0.0001$ ($n = 3$ replicates); one-way ANOVA. **h** Soft agar colony formation of immortal Hras transformed MEFs. Colonies are grown in the absence of Dox for 20 days (gray upper panels), or in the presence of Dox since plating (purple mid panels). Additionally, colonies grown in the absence of Dox were supplemented with Dox at day 8 (lower panels). Colony diameter (in microns) is quantified in the right histogram. Each dot represents one colony (over 50 colonies are quantified in each set). *$p < 0.01$; ****$p < 0.0001$, one-way ANOVA Bonferroni test

lagging chromosomes and anaphase bridges (Fig. 3a–e) resulting in increased duration of mitosis (Fig. 3d). In line with these defects, a significant percentage of cells (26.7% vs. 1.1% in control cells; Fig. 3a, b) exited mitosis in the absence of chromosome segregation. In addition, 38% of Plk1-overexpressing cells displayed abnormal cytokinesis resulting in binucleated cells or

underwent mitotic regression (7.0% of Dox-treated vs. 3.4% untreated cells) thereby generating tetraploid cells with a single nucleus (Fig. 3a, b). Immunofluorescent analysis of these cultures revealed increased mitotic aberrations such as lagging chromosomes and cytokinesis bridges (Fig. 3e). Finally, treatment of these cultures with Dox resulted in the formation of a significant

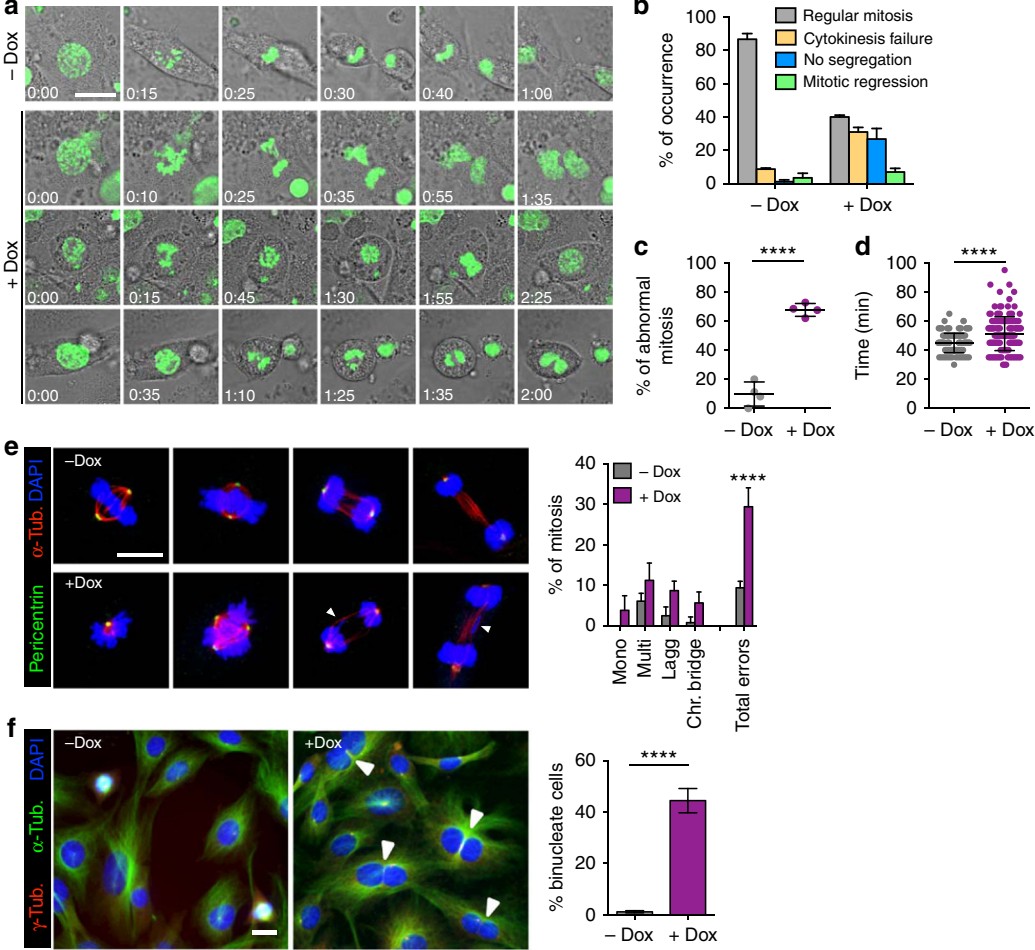

**Fig. 3** Mitotic defects in Plk1-overexpressing MEFs. **a** Time-lapse microscopy of Plk1 MEFs untreated (upper panel) and after 8 h on Dox (lower panels), indicating mitotic cells; H2B-GFP (green), classified based on the three major phenotypes resulting in a tetraploid progeny in the case of Plk1 overexpression. Scale bar 20 μm. **b** Percentage of occurrence of each major phenotype in –Dox ($n = 141$) and +Dox ($n = 164$) MEFs. **c** Percentage of mitotic errors per MEF (–Dox: 141 cells; +Dox: 164 cells); points represent individual MEF line. **d** Duration of mitosis in the MEF cultures (–Dox: 141 cells; +Dox: 164 cells). In **c**, **d**, ****$p < 0.0001$, Student's $t$-test. **e** Immunofluorescence against α-tubulin and pericentrin in primary MEFs after 24 h on Dox followed by quantification of each mitotic phase and percentage of mitotic aberrancies (–Dox: $n = 149$ mitotic cells; +Dox: $n = 166$ mitotic cells). ****$p < 0.0001$; two-way ANOVA. Scale bar, 10 μm. **f** Immunofluorescence against α-tubulin and γ-tubulin in (+/Plk1)(rtTA/rtTA) MEFs untreated (–Dox) or treated with Dox for 48 h (+Dox). Binucleated cells (arrowheads) from the total population (>100 cells from at least four random microscope fields are quantified in each replicate). ****$p < 0.0001$ ($n = 2$ replicates); one-way ANOVA. Scale bar, 10 μm

number of binucleated cells (Fig. 3f), in agreement with a failure in cytokinesis.

To determine whether elevated Plk1 levels resulted in increased Plk1 activity at different cellular localizations, we stained Plk1-overexpressing cells with an antibody that recognizes phosphorylation on Thr210 at the Plk1 T-activation loop. Quantification of this signal revealed a significant increase in Plk1-pT210 levels in interphase cells with centrosomes separated into the two poles (characteristic of late G2 cells; Fig. 4a). Furthermore, this signal was also significantly increased in prometaphase and metaphase cells (Fig. 4b), as well as in single prometaphase kinetochores during prometaphase in Plk1-overexpressing cells (Fig. 4c). The centromeric levels of Sgo1, a protector of centromeric cohesion, were reduced in Plk1-overexpressing cells (Fig. 4d) in agreement with previous data suggesting that Plk1 activity leads to dissociation of Sgo1 from centromeres[24]. In line with these observations, a significant number of Plk1-overexpressing cells displayed reduced cohesion as observed in metaphase spreads after treatment of cells with the microtubule poison colcemid (Fig. 4e).

**Plk1 overexpression impairs cell abscission.** Cytokinesis failure was among the most common phenotypes induced by Plk1 overexpression leading to 40% of binucleated cells in cultured MEFs (Fig. 3f). Plk1 is known to modulate cell abscission by regulating the recruitment of endosomal sorting complex required for transport (ESCRT) machinery to the site of bridge severing[25–27]. During mitosis, Plk1 phosphorylates Cep55 thereby preventing its premature association with the midzone until cytokinesis entry[25,26]. Plk1 inactivation during anaphase allows Cep55 dephosphorylation and translocation to the midbody, thereby resulting in the localization of ESCRT proteins to the bridge. We therefore asked whether Plk1 overexpression could lead to defective abscission as a consequence of deregulation of this pathway.

A detailed analysis of cytokinesis defects in Plk1-overexpressing cells showed a significant number of cytokinesis aberrations (Fig. 5a) as well as increased length of the cytokinesis bridge in agreement with delayed abscission (Fig. 5b). During cytokinesis, the midbody is formed by compacted bundles of microtubules and proteins required for abscission such as

RacGAP1 (also known as Cyk4) and MKLP1 (also known as Kif23 or centraspindlin). RacGAP1, a Plk1 substrate involved in RhoA activation and local actomyosin contraction[28,29], was properly activated at the midbody in Plk1-overexpressing cells (Supplementary Fig. 5a). Similarly, loading of the midbody core kinesin MKLP1, a plus-end directed motor protein involved in the formation of the cleavage furrow in late anaphase and in cytokinesis[30], was not affected (Supplementary Fig. 5b), suggesting no defects in the preassembled midbody in cells over-expressing Plk1.

Abscission is ultimately mediated by binding of the adapter protein Cep55 to MKLP1 in the preassembled midbody. During mitosis, phosphorylation of Cep55 by Plk1 prevents its premature loading to this structure, whereas Plk1 degradation during anaphase results in Cep55 dephosphorylation and loading to the midbody[26]. Cep55 then promotes abscission by recruiting ESCRT-I membrane-remodeling proteins[27,31,32]. In agreement with a role for Plk1 in preventing Cep55 loading, Plk1 overexpression resulted in a significant reduction in the localization of Cep55 to the midbody (Fig. 5c). Mislocalization of Cep55 was accompanied by reduced loading of the ESCRT-I component TSG101 (Fig. 5d). Importantly, direct inhibition of Plk1 kinase activity by 1-h treatment with the Plk1 inhibitor BI2536 significantly rescued the midbody levels of Cep55 (Fig. 5c) and TSG101 (Fig. 5d) in Plk1-overexpressing cells. Moreover, short treatment with the Plk1 inhibitor resulted in lower rates of

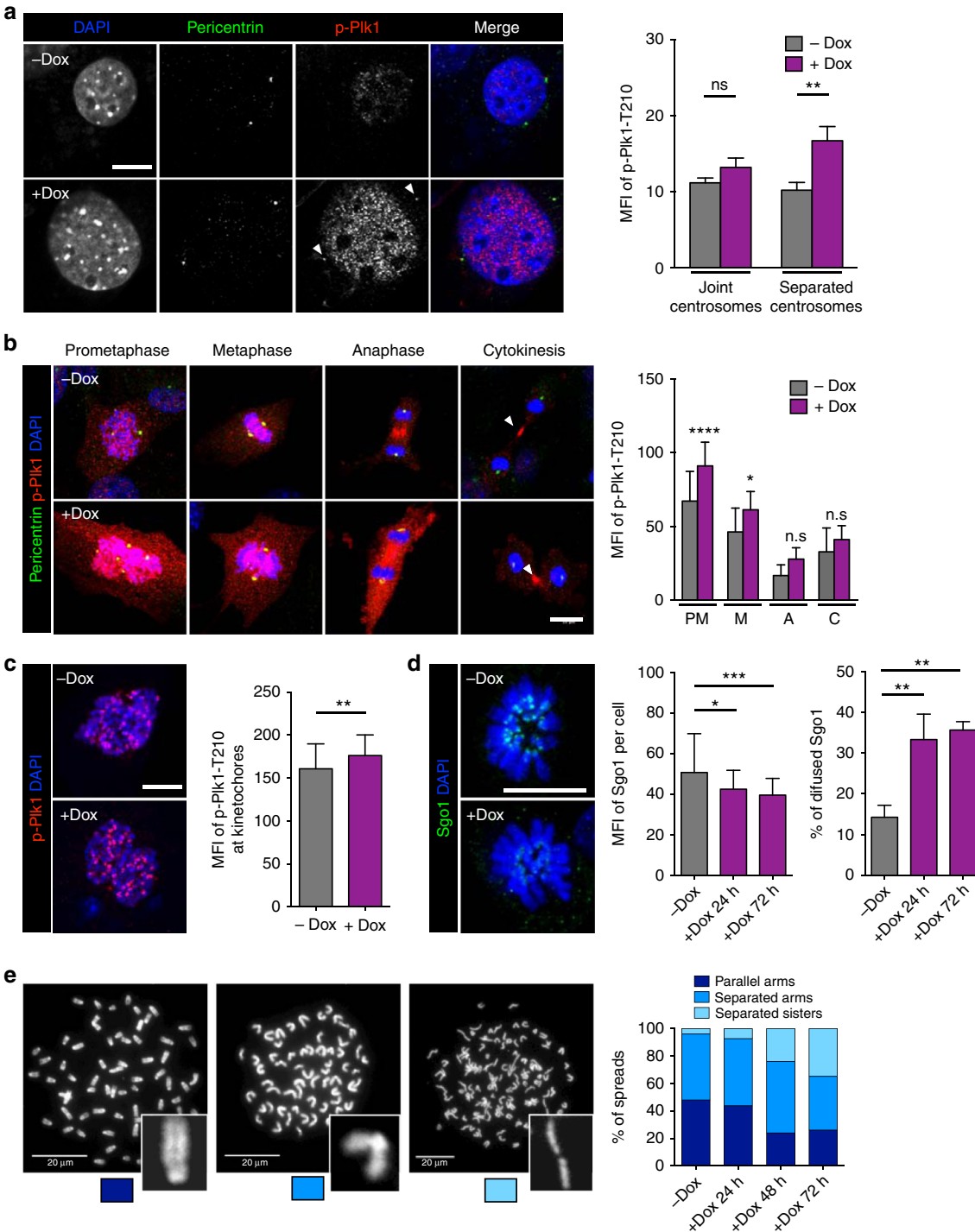

binucleation (Fig. 5e), suggesting that cytokinesis failure is a consequence of increased kinase activity and defective midbody loading of Cep55 and ESCRT complexes in Plk1-overexpressing cells.

**Delayed tumor progression in Plk1-overexpressing mice.** To monitor the effects of Plk1 overexpression during tumor development in vivo, we used a mouse mammary gland tumor model in which an oncogenic *Kras* (*Kras*[G12D]) allele is induced by a tetracycline transactivator expressed under the control of the mammary gland tumor virus (MMTV) sequences[5]. These mice develop mammary gland tumors with complete penetrance and a median latency of 23 weeks. Strikingly, Plk1 overexpression prevented tumor development by Kras[G12D] in 85% of the mouse colony and significantly delayed tumor latency in the rest of the colony (Fig. 6a). We reproduced these results using a second model in which breast tumors were induced by the Her2 oncogene[5]. Similar to the previous model, Plk1 overexpression also prevented tumor development in 45% of the colony and significantly delayed tumor onset in the remaining animals (Fig. 6b). In addition, the number of tumors per animal was significantly reduced after Plk1 overexpression in these models (Fig. 6c). In line with the polyploid phenotype observed in cultured MEFs, overexpression of Plk1 resulted in increased nuclear volume in tumor cells (Supplementary Fig. 6a, b). In addition, these Plk1-overexpressing tumor cells displayed a higher frequency of aneuploidy and polyploidy compared to the single oncogene tumors (Fig. 6d–f). To analyze if high Plk1 levels influence chromosome instability during tumor growth, we performed time lapse microscopy of tumor cells. Indeed, 50% of *Her2/Plk*1 tumor cells displayed mitotic errors compared to 24% of Her2 alone. In addition, 25% of *Her2/Plk*1 tumor cells became polyploid while this only occurred in 3% of *Her2* tumor cells (Fig. 6g, h). A common consequence of polyploidy is the activation of p21 through a p53-dependent mechanism resulting in cell cycle arrest. In agreement with our previous data in Plk1-overexpressing MEFs, we also observed a significant increase of p21-positive cells in Plk1-expressing tumors, compared to oncogene alone (Supplementary Fig. 6c, d). Moreover, the percentage of cells proliferating in these tumors was significantly lower than in *Kras* or *Her2* tumors alone (Supplementary Fig. 6c, d), suggesting that in mammary tumors Plk1 overexpression results in increased CIN and growth inhibition.

To identify early effects of Plk1 overexpression in vivo, we surgically removed mammary glands from the *Kras*[G12D]; *Plk*1 model 4 days and 100 days after induction of the transgenes (Supplementary Fig. 7a, b). Immunostaining with a Plk1 antibody as well as RT-PCR confirmed the expression of both *Kras* and *Plk*1 transgenes (Supplementary Fig. 7a, b). Kras[G12D] and *Plk*1-overexpressing models displayed increased number of phospho-histone H3-positive cells suggesting either increased cell proliferation or increased duration of mitosis (Supplementary Fig. 7c). However, Plk1-overexpressing tissues were characterized by the presence of cells with higher nuclear volume (Supplementary Fig. 7d) as well as increased apoptotic cell death with independence of Kras status (Supplementary Fig. 7e) at these early time points, likely contributing to reduced tumor burden in the presence of elevated Plk1 levels.

To monitor single-cell fate of epithelial cells in vivo after Plk1 overexpression, we used a three-dimensional culture system of primary mammary epithelial cells[5,33] isolated from these mouse models. Single cells were embedded in matrigel and allowed to develop acinar structures for 6–8 days. Once spheres were formed, we induced transgene expression with Dox and followed cell division of these cells co-expressing GFP-tagged histone H2B using time-lapse microscopy. In line with previous results[5], control and *Kras*[G12D]-induced cultures had no obvious phenotype after 36 h on Dox (Fig. 7a) and the mitosis observed were normal. However, similar to the observations in cultured MEFs, overexpression of Plk1 in mammary epithelial cells resulted in a significant prolonged mitosis (Fig. 7a, b) in the presence of abnormal mitotic figures, including lagging chromosomes, defective chromosome segregation, and cytokinesis failure (Fig. 7c, d). All together, these data suggest that Plk1 overexpression results in defective mitosis and cytokinesis in mammary gland epithelial cells, thus suppressing tumor development.

**Plk1 expression correlates with genome-doubled breast cancers.** Next, to explore the relationship between PLK1 expression and ploidy in human breast cancers, we obtained matched copy number and expression data for 953 breast tumors from the TCGA. Consistent with an association between *PLK1* expression and polyploidy, we observed a highly significant difference in levels of *PLK1* expression in tumors which exhibited evidence of having undergone a genome doubling event during their evolution compared to their non-doubled counterparts (Fig. 8a, $p = 4.62e-09$, *t*-test). Furthermore, the association between genome doubling and *PLK1* expression levels remained significant in *TP53* wild-type tumors, suggesting that *PLK1* may facilitate polyploidization in a *TP53*-independent manner (Fig. 8b, $p = 3.37e-06$, *t*-test).

Finally, to investigate the clinical significance of *PLK1* expression in breast cancer, we grouped tumors into those with low *PLK1* expression (bottom quartile *PLK1* expression) and

**Fig. 4** Plk1 overexpression induces defects in centrosome and chromosome segregation dynamics. **a** Immunofluorescence against p-Plk1-T210 and pericentrin in MEFs untreated or after 24 h with doxycycline (+Dox). The histogram shows the quantification of mean fluorescence intensity (MFI) of p-Plk1-T210 staining at the centrosomes during late G2 phase (separated; –Dox: $n = 54$ centrosomes; +Dox: $n = 34$ centrosomes) or earlier interphase (joint; –Dox: $n = 88$ centrosomes; +Dox: $n = 58$ centrosomes; $n = 2$ replicates). Scale bar, 10 μm. **b** Immunofluorescence against p-Plk1-T210 and pericentrin in MEFs untreated or after 24 h on Dox. The histogram shows the quantification of mean fluorescence intensity (MFI) of p-Plk1-T210 staining at each mitotic phase [Prometaphase (PM): –Dox $n = 22$ cells; +Dox: $n = 26$ cells; Metaphase (M): –Dox $n = 17$ cells; +Dox: $n = 21$ cells; Anaphase (A): –Dox $n = 15$ cells; +Dox: $n = 12$ cells; Cytokinesis (C): –Dox $n = 13$ cells; +Dox: $n = 15$ cells; $n = 3$ replicates]. **c** Immunofluorescence against p-Plk1-T210 in primary MEFs untreated or after 24 h on Dox. The histogram shows the quantification of mean fluorescence intensity (MFI) of p-Plk1-T210 staining at individual kinetochores of cells in prometaphase (–Dox: $n = 51$ kinetochores; +Dox: $n = 51$ kinetochores; $n = 2$ replicates) **$p < 0.01$; Mann–Whitney test. Scale bar, 10 μm. **d** Immunofluorescence against Sgo1 in primary MEFs untreated or after 24 and 72 h on Dox. The first histogram shows the quantification of mean fluorescence intensity (MFI) of Sgo1 staining in the entire cell at prometaphase (–Dox $n = 33$ cells; +Dox 24 h: $n = 35$ cells; +Dox 72 h: $n = 52$ cells; $n = 3$ replicates). The second histogram shows the percentage of prometaphase cells with diffused Sgo1 staining ($n = 3$ replicates). Scale bar, 10 μm. In **a**, **b**, **d**, n. s. not significant; *$p < 0.05$; **$p < 0.01$; ***$p < 0.001$; ****$p < 0.0001$; one-way ANOVA. **e** Chromosome spreads (DAPI stained) from (+/*Plk*1);(rtTA/rtTA) MEFs untreated or treated with Dox at the indicated times. Chromosome cohesion was classified in three different status: "parallel arms" as readout of full chromatid cohesion (dark blue box), "separated arms" as readout of chromatid arm separation (mid blue), and "separated sisters" as readout of fully separated chromatids (light blue). (–Dox, $n = 24$; +Dox 24 h, $n = 40$; +Dox 48 h, $n = 25$; +Dox 72 h, $n = 22$ cells). Scale bars, 20 μm

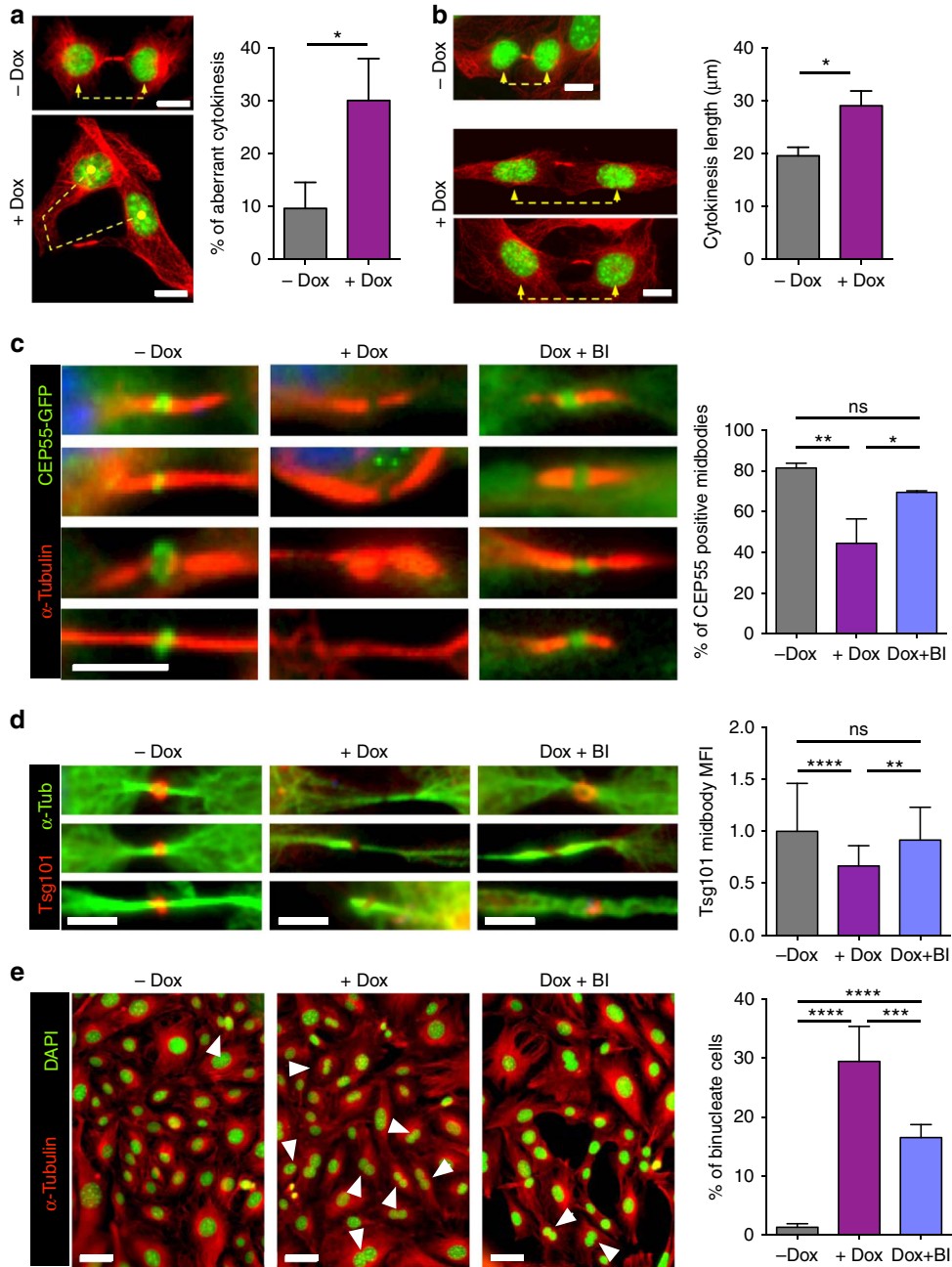

**Fig. 5** Overexpression of Plk1 impairs Cep55 and ESCRT loading into the cytokinesis midbody. **a** (+/Plk1);(rtTA/rtTA) MEFs were untreated (–Dox) or treated for 24 h with doxycycline (+Dox), fixed and stained for α-tubulin (red) and DAPI (DNA, green). Cells in cytokinesis (n > 100 per condition; n = 3 replicates) were evaluated for aberrant cytokinesis, considering the midbody formation and shape, and correct distribution of DNA into the two daughter cells. Scale bars, 10 μm. **b** The length of the cytokinesis bridge was evaluated by measuring the distance in between the two daughter nuclei in MEFs untreated or after 24 h of Dox treatment (–Dox, n = 17 cells; +Dox, 23 cells). Scale bars, 10 μm. In **a**, **b** *, p < 0.05; Student's t-test. **c** MEFs expressing a CEP55-EGFP fusion (green) were treated for 24 h with Dox (+Dox), in the absence or presence of 1 μM of the Plk1 inhibitor BI2536 for 1 h at the end of the Dox time (Dox + BI), or left untreated as control (–Dox). α-tubulin is in red. Data represent the percentage of cells with positive CEP55-EGFP signal at the midbody (–Dox, n = 134; +Dox, n = 94; Dox + BI, n = 47 cells; n = 3 replicates (–Dox, +Dox) or 2 replicates (Dox+BI)). Scale bar, 5 μm. **d** MEFs expressing a Tsg101-mCherry fusion (red) were treated for 24 h with Dox (+Dox), in the absence or presence of 1 μM of BI2536 for 1 h at the end of the Dox time (Dox + BI), or left untreated as control (–Dox). α-tubulin is in green. Data represent Tsg101-mCherry mean intensity at the midbody (–Dox, n = 81; +Dox, n = 60; Dox + BI, n = 37 cells; n = 2 replicates). Scale bars, 5 μm. **e** MEFs were treated for 24 h with Dox (+Dox), in the absence or presence of 1 μM of BI2536 for 1 h at the end of the Dox time (Dox + BI), or left untreated (–Dox). Cells were stained for α-tubulin (red) and DAPI (DNA, green) and binucleation index was quantified from more than 600 cells in each sample (n = 5 replicates). Scale bars, 50 μm. In **c**–**e**, n.s. not significant; *p < 0.05; **p < 0.01; ***p < 0.001; ****p < 0.0001; one-way ANOVA

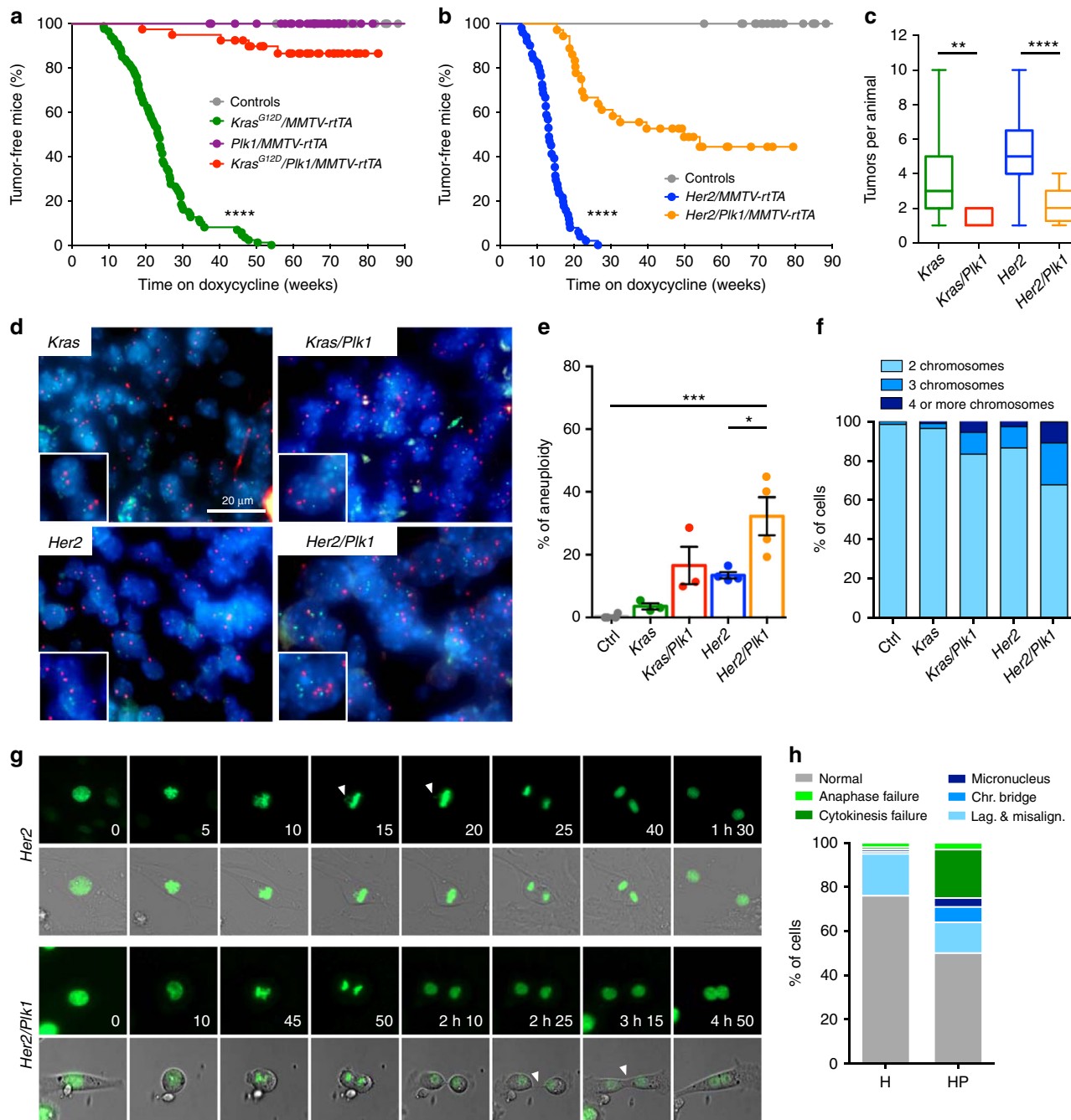

**Fig. 6** Plk1 overexpression reduces tumor development induced by Kras or Her2 oncogenes. **a** Percentage of tumor-free mice after doxycycline administration (control, n = 33; Plk1/MMTV-rtTA n = 35; Kras$^{G12D}$ n = 87; Kras$^{G12D}$/Plk1 n = 40 mice). **b** Percentage of tumor-free survival after doxycycline administration (control, n = 33; Her2, n = 51; Her2/Plk1, n = 36 mice). In **a**, **b**, ****p < 0.0001, Mantel–Cox test. **c** Number of tumors per animal in the indicated genotypes; **p < 0.01; ****p < 0.0001; Mann–Whitney test. **d** Interphase-Fluorescent in situ hybridization (I-FISH) on paraffin sections of mammary tumors using two centromeric probes against chromosomes 16 and 17. **e** Quantification of the frequency of aneuploidy (left) in samples from the indicated genotypes (control, n = 4; Kras, n = 4; Kras/Plk1, n = 3; Her2, n = 4; Her2/Plk1, n = 4 mice). *p < 0.05; ***p < 0.001; one-way ANOVA. **f** Percentage of cells with 2, 3, or 4 or more chromosomes in the indicated genotypes. **g** Representative micrographs of Her2 and Her2/Plk1 tumor cells in vitro (H2B-GFP green). Top: mitotic cell with a lagging chromosome. Bottom: cytokinesis failure resulting in binucleation. **h** Percentage of cells in Her2 tumors (H) and Her2/Plk1 (HP) with the indicated mitotic errors

those with higher *PLK1* expression (remaining quartiles). In keeping with the observed tumor suppressor properties of *PLK1*, patients with low *PLK1* expression were associated with significantly shorter overall survival, compared to those with higher *PLK1* expression levels (Fig. 8c, p = 0.0099; hazard ratio (HR), 0.59; 95% confidence interval (CI), 0.40–0.89). These results remained significant in multivariate analysis, including

PAM50 subtypes, genome doubling and *TP53* status in the model (p = 0.00144, HR = 0.51, CI = 0.33–0.77).

## Discussion

Plk1 belongs to a family of kinases with multiple roles in proliferative as well as postmitotic cells[10–13,22]. Among Polo-like

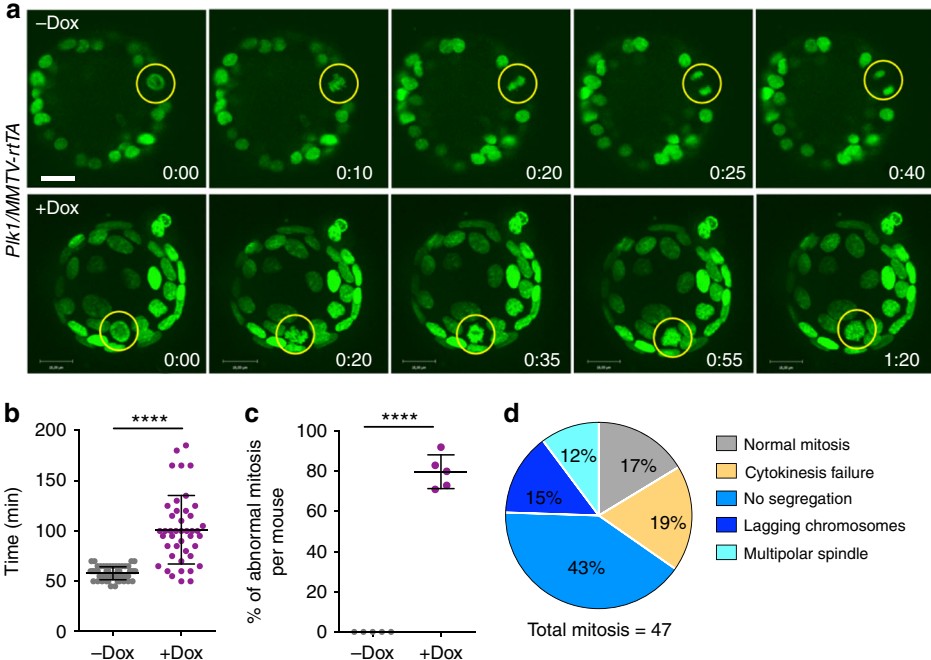

**Fig. 7** Mitotic aberrations in cultured Plk1-overexpressing mammary organoids. **a** Time-lapse microscopy of *Plk1/MMTV-rtTA* mammary organoids expressing H2B-GFP either untreated (upper panel) or after 24 h on Dox (lower panel). Yellow circles indicate mitotic cells; H2B-GFP (green). Scale bar, 18 μm. **b** Duration of mitosis in the organoid cultures (−Dox, 57 cells; +Dox, 43 cells). **c** Percentage of mitotic errors per mouse (−Dox, 57 cells from 5 mice; +Dox, 47 cells from 5 mice). In **b**, **c**, ****$p < 0.0001$, Student's *t*-test. **d** Classification of mitotic phenotypes in organoid cells overexpressing Plk1 after 36 h on Dox

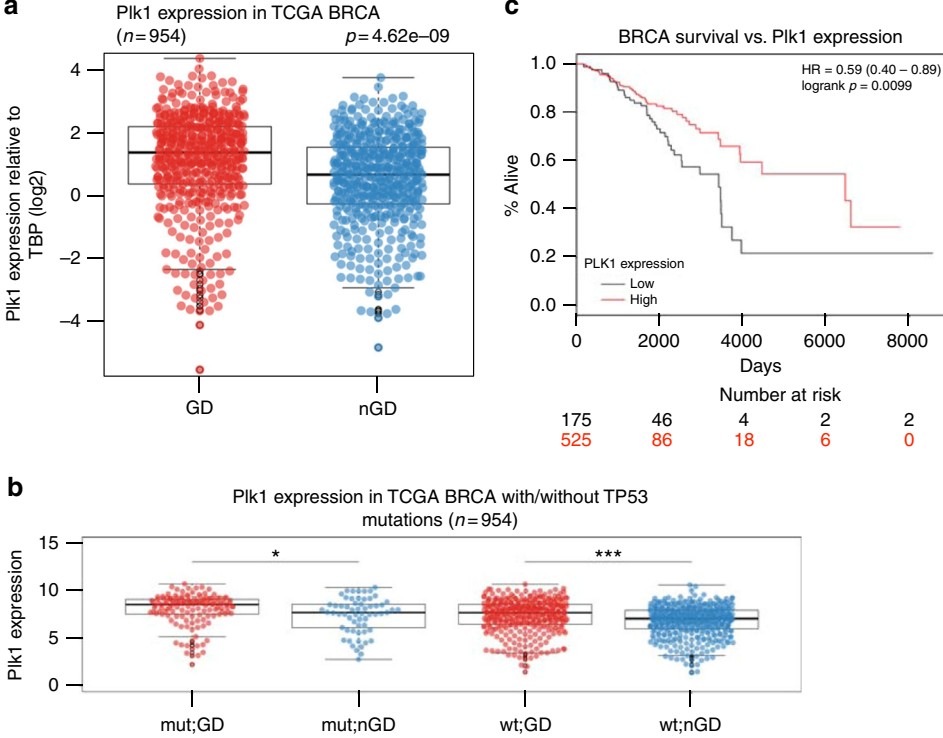

**Fig. 8** Plk1 expression in human breast cancers. **a** Plk1 expression relative to TBP (log2) in genome-doubled (GD) and non-genome-doubled (nGD) breast cancers from the TCGA. $p = 4.62e-09$, *t*-test. **b** Plk1 expression in breast cancers with and without TP53 mutations. The association between GD and Plk1 expression is significant in TP53 wt tumors. $p = 3.37e-06$, *t*-test. **c** Survival analysis of breast cancer (BRCA) patients with low Plk1 expression (bottom quartile) and those with higher Plk1 expression (remaining quartile) $p = 0.0099$; hazard ratio (HR), 0.59; 95% confidence interval (CI), 0.40–0.89

kinases, Plk1 is considered an interesting target for cancer therapy due to the requirements for its kinase activity during cell division in tumor cells[13,20]. However, to what extent Plk1 expression is a cause or a consequence of carcinogenesis is under debate[34]. Plk1 expression is cell cycle-dependent and Plk1 is frequently overexpressed together with other mitotic genes in highly proliferating and chromosomally unstable tumors[35]. Plk1 is repressed by the tumor suppressor p53[36] and increased Plk1 levels can simply reflect p53 inactivation in cancer cells[37].

Although Plk1 is frequently classified as an oncogenic protein, its contribution to malignant transformation is arguable. In pioneer assays in the 90's, ectopic expression of Plk was shown to increase DNA synthesis and mitosis in quiescent NIH 3T3 cells and to induce oncogenic foci[38]. Although the molecular basis for these observations is unknown, Plk1 can inhibit p53[39] and leads to the stabilization of Myc proteins[40], suggesting possible mechanisms for proliferative functions. However, its sufficiency in triggering cell cycle entry and progression is unclear and more recent data suggest that Plk1 overexpression leads to cell proliferation defects at least partially due to aberrant mitosis and the activation of the spindle assembly checkpoint[41]. Interestingly, there are early reports showing that other Polo like family members might also play as tumor suppressors such as Plk3 or Plk5[10,42]. Recent data also suggests that constitutive overexpression of Plk1 in mice does not lead to tumor formation[43], although the mechanism has not been investigated. In our hands, overexpression of Plk1 using an inducible knockin model in murine cells results in decreased cell proliferation accompanied by multiple mitotic aberrations, including defects in chromosome congression and segregation, and abnormal cytokinesis (Figs. 2–5). Plk1 overexpression also prevents malignant transformation of primary cells by Ras oncogenes (Fig. 2) and impairs breast cancer development induced by Kras or Her2 oncogenes (Fig. 6).

The pleiotropic defects caused by Plk1 overexpression are likely a consequence of the multiple roles of this protein in several cellular structures such as the centrosomes, kinetochores, or the cytokinesis bridge[12,13,22]. Among these defects, failure in cytokinesis and abscission is the most abundant defect in Plk1-overexpressing cells, resulting in the formation of binucleated cells as well as tetraploid mononucleated cells generated after mitotic regression in the absence of abscission (Fig. 3). Lack of abscission in these cells correlates with defective loading of Cep55, a Plk1 substrate that recruits the ESCRT component TSG101 to the cytokinesis bridge[26]. In the presence of high Plk1 activity, loading of the ESCRT complex into the cytokinesis bridge is deficient (Fig. 5), likely as a consequence of impaired loading of Cep55 during late stages of anaphase. In fact, treatment of Plk1-overexpressing cells with the Plk1 inhibitor BI2536 partially rescued these defects, suggesting that cytokinesis defects in Plk1-overexpressing cells are kinase-dependent and at least partially mediated by the Cep55-ESCRT pathway. Since Plk1 is also involved in cell migration and metastasis[44], authophagy[45], pentose phosphate metabolism[46] or blood pressure regulation[11], overexpression of this kinase in cancer cells may result in a variety of additional abnormalities that will deserve future analysis.

Accumulative evidences in the last decade suggest that either downregulation or upregulation of several mitotic regulators induce genomic instability. As a pioneering example, both the downregulation[47] and overexpression[6] of Mad2 result in CIN accompanied by the formation of chromosomally unstable tumors in vivo. Other examples in which overexpression of a mitotic regulator is accompanied of CIN and tumor formation include Aurora A[48,49], Aurora B[8], and Cyclin B1 and Cyclin B2[7], among others[2,9]. Contrary to these examples, Plk1 overexpression results in a significant delay and decreased breast cancer

incidence in combination with Kras and Her2 oncogenes. These results are in agreement with previous observations suggesting a tumor suppressor role for Plk1. For instance, Plk1(+/−) mice display increased susceptibility to spontaneous tumor development[50]. Although Plk1 mutations are not frequent in human cancer cells, the only mutations found in a few cancer cell lines generate Plk1 isoforms with decreased protein stability and reduced protein levels[51]. In human breast cancer cells, Plk1 mediates estrogen receptor (ER)-regulated gene transcription participating in the expression of genes involved in developmental and tumor-suppressive functions[52]. Finally, contrary to other studies, reduced levels of PLK1 have been correlated to aggressiveness, decreased response to chemotherapy, and poor prognosis for disease-free survival in colorectal cancer[53]. Similarly, PLK1 protein levels correlate with better prognosis in p53 wild-type breast tumors[54]. A meta-analysis of breast cancer patients suggests that PLK1 overexpression at the mRNA level correlates with poor prognosis in ER+. However, PLK1 overexpression correlates with better prognosis in ER-negative and Her2-positive patients (Supplementary Fig. 8). Our analysis of human breast cancer datasets shows that tumors with higher PLK1 expression generally have improved prognosis (Fig. 8). All together, these data indicate that, despite being generally considered as an oncogene, PLK1 may have tumor-suppressive activities.

The fact that Plk1 may function as a tumor suppressor instead of an oncogene does not necessarily argue against the use of Plk1 inhibitors in cancer therapy. Many essential components of cell proliferation may be used as cancer targets despite having no oncogenic activity, owed to non-oncogene addiction of cancer cells for specific cellular processes such as cell division. Plk1 is a frequent hit in chemical or genome-wide genetic screens to uncover new targets under different oncogenic backgrounds[55–64]. In fact, Plk1 inhibition may be particularly effective in Ras-induced[55,65] or Her2-induced[66] tumors indicating that, independently of a possible tumor suppressor role of overexpressed Plk1, efficient kinase inhibition of this protein may impair tumor cell proliferation and survival. Plk1 inhibitors are currently being tested in multiple solid and hematopoietic tumors and BI6727 (volasertib), a derivative of BI2536, has recently received the FDA Breakthrough Therapy designation for its effect in acute myeloid leukemia[20,67]. Understanding the specific requirements for Plk1 as compared to other essential cell cycle kinases will likely contribute to a better design of therapeutic strategies against its kinase activity.

## Methods

**Mouse models.** KH2 ES cells, kindly provided by Konrad Hochedlinger and Rudolf Jaenisch[21], carried the M2-rtTA gene inserted within the Rosa26 allele. A cassette containing the FLAG-human Plk1 cDNA under the control of the Dox-responsive promoter (tetO) was inserted downstream of the Col1A1 locus (Fig. 1a). ColA1-Plk1 heterozygous animals were bred to MMTV-rtTA, TetO-KrasG12D, TetO-rat-Her2, and H2B-GFP[5]. Only heterozygous female animals for all transgenes were used in this study. Breeding and experimentation was performed at the CNIO, EMBL-Monterotondo, and DKFZ animal facilities, with ethical approval from the corresponding Animal Welfare and Ethical Review Bodies and national and European legislations.

Mice with different combinations of the Plk1;rtTA alleles [(+/+)(rtTA/rtTA); (+/Plk1)(+/rtTA); (+/Plk1)(rtTA/rtTA); (Plk1/Plk1)(rtTA/rtTA)] were fed with 2000 ppm Dox impregnated food in order to achieve ubiquitous transgene expression and evaluate Plk1 expression tolerance. In all other in vivo experiments mice were fed with 625 ppm Dox-enriched diet to exclusively express the transgenes in the mammary gland.

Quantitative analysis of blood cell populations was performed in mice after 5 days of 2000 ppm Dox food administration. Blood was retrieved in tubes supplemented with K3-EDTA 3 K as an anticoagulant (Aquisel; 1501126) and the analysis was performed with a Procount veterinary hematology analyzer (serial number 901235).

Isolation of tail-DNA was performed via incubation in 200 μL 0.05 M NaOH at 98 °C for 1.5 h and subsequent neutralization with 20 μL 1 M Tris HCl pH 7.5.

*TetO-Kras*, *TetO-Her2*, and *MMTV-rtTA* transgenic mice were genotyped as described previously[5]. The following oligonucleotides were used to genotype the ColA1-Plk1 allele: KH2-Plk1 A: 5′-GCACAGCATTGCGGACATGC-3′, KH2-Plk1 B: 5′-CCCTCCATGTGTGACCAAGG-3′, KH2-Plk1 C: 5′-GCAGAAGCGCGGCCGTCTGG-3′. For all transgenes, the following PCR program was applied: 94 °C for 2 min, 30 times [95 °C for 30 s, 60 °C for 30 s, 72 °C for 30 s], and a final step at 72 °C for 1 min.

**Immunodetection in tissue sections**. Immunohistochemistry and immunofluorescence in mouse tissues was performed using formalin-fixed paraffin-embedded sections. Following deparaffinization with xylene and rehydration with graded ethanol, antigen retrieval was performed using 0.09% (v/v) unmasking solution (Vector Labs) for 30 min in a steamer. Inactivation of endogenous peroxidases was carried out using 3% Hydrogen Peroxide (Sigma) for 10 min. Secondary antibody staining and biotin-streptavidin incubation were performed using species-specific VECTASTAIN Elite ABC kits (Vector Labs). DAB Peroxidase Substrate kit (Vector Labs) was utilized for antibody detection. Eosin Y and haematoxylin were from Bio-Optica and Vector Labs. Primary antibodies used were anti-pH3 Ser10 (1:200, Cell Signaling, 9701), FLAG (1:200, Sigma), Plk1 (1:20)[68], p21 (1:50, SC-6246), and PCNA (1:8000, AB18197). 10% goat serum (Jackson Immuno) and species-specific Alexa fluorophore-labeled goat IgG (1:800, Invitrogen) were used as secondary antibodies. TUNEL (In Situ Cell Death Detection Kit, TMR red, Roche, #12156792910) was used for detecting cell death. Analysis of images was performed using Fiji (https://fiji.sc/). Tumor sections were visualized under a TissueFAXS slide scanning platform (TissueGnostics, Vienna, Austria). For the PCNA, p21, and nuclear size analysis, the quantitation was performed using StrataQuest software (TissueGnostics) to determine the percentage of PCNA$^+$, p21$^+$ cells, and/or nuclear area.

Interphase-FISH was performed on formalin-fixed paraffin-embedded 5-μm sections. Following deparaffinization with xylene and rehydration with graded ethanol, Vysis Paraffin Pretreatment IV and Post-Hybridization Wash buffer kit was used as specified by the manufacturer. Probe mix was prepared with 3 μL of labeled probe and 7 μL Vysis LSI/WCP Hybridization buffer. Hybridization was performed using Abbott Molecular Thermobrite system with the following program: Denaturation 76 °C for 5 min, hybridization at 37 °C for 20–24 h. Pan-centromeric probes were made using pairs of BAC clones for each chromosome. Chr 16-RP23–290E4 and RP23–356A24 labeled with SpectrumOrange-dUTP (Vysis), and Chr 17-RP23–354J18 and RP23–202G20 labeled with SpectrumGreen-dUTP (Vysis); the BAC DNAs were labeled by nick translation according to standard procedures. Signal for hybridization for each probe was checked in a minimum of 60 interphase cells as reported previously[6].

**Cell culture and immunofluorescence**. MEFs were prepared from E13.5 embryos using standard procedures[6]. Primary MEFs were immortalized by using the T121 construct that encodes the first 121 amino acids of the SV40 large T antigen. MEFs growth curves were done by measuring cell culture confluence (JuLI™ FL, Nanoentek), each 24 h after doxycycline addition. Doxycycline concentration in all culture experiments was 1 μg/ml. Cell cycle profiling analysis was performed by counterstaining DNA with 4,6 diaminophenylindole (DAPI) and using the FACS-Canto flow cytometry device (BD Biosciences). Apoptosis was determined by Annexin V-FITC staining (BD Pharmingen™) using the FACS-Canto flow cytometry device (BD Biosciences). For MEF proliferation analysis, EdU was added to exponential growing MEFs for 1 h, and then cells were tripsinized and fixed in cold 70% ethanol. EdU staining protocol was done following manufacture instructions (Click-iT, Invitrogen), and DNA was stained with DAPI for 30 min. Entry in S-phase was quantified by synchronizing MEFs in G0 by confluency and serum starvation (FBS 0.1%) during 48–72 h. Then, MEFs were split and seeded in 10% FBS. S-phase entry was analyzed by EdU incorporation at different time points after cell seeding (up to 32 h). Mitotic index was determined by immunostaining with anti-histone H3-phosphoSer10 (1:500, Millipore 06–570). Cell senescence was measured by the senescence-associated beta-gal staining kit (Cell Signaling).

Replicative stress was monitored by cell immunostaining on μCLEAR bottom 96-well plates (Greiner Bio-One), with antibodies against γ-H2aX (1:1000, Millipore 05–636) and 53BP1 (1:1000, Novus Biologicals NB100–304). DNA was counterstained with DAPI. Images were acquired and quantify by an Opera High-Content Screening System (PerkinElmer). A 20× magnification lens was used and pictures were taken at non-saturating conditions.

For chromosome cohesion analysis, MEFs were arrested in 0.1 μg/ml colcemid (Karyomax; Thermo Fisher) for 4–6 h, harvested by trypsinization, swollen in 75 mM KCl for 30 min at 37 °C and fixed in methanol:acetic acid 3:1. Fixed cells were then dropped onto slides to obtain chromosome spreads that were stained and mounted with DAPI. Images were taken under a Leica DM6000 microscope.

H-Ras transformation in MEFs was achieved by transfecting immortal MEFs with HRas-V12 or a combination of HRas-V12 and E1A in primary MEFs. Cells were then allowed to grow until contact inhibition, and then maintained in culture for at least 3 weeks with fresh media changes every 48 h. For focus assays, doxycycline was added to the cells once they reached confluency, and before transformed foci were present. After 3 weeks, cell transformation colonies were either stained with Giemsa and number of colonies quantified using ImageJ software or individual colonies grown in the absence of Dox were expanded for

further experiments. For anchorage-independent growth analysis, HRas-transformed MEF colonies with different Plk1;rtTA genotypes were seeded in soft agar media using routine procedures.

DNA transfection in MEFs was done using the Lipofectamine 3000 kit (Invitrogen). MEFs were transfected either with cDNA encoding for MKLP1 fused to GFP (Addgene #70145), CEP55 (obtained from the Mammalian Gene Collection) fused to GFP, or the ESCRT-I member TSG101 fused to the mCherry reporter (Addgene #38318). 48 h after transfection, the expression of each cDNA was evaluated by immunofluorescence. Time-lapse imaging was performed for 12 h using a Leica SP5 Confocal microscope: 2 μm optical sectioning across 12 μm stack, every 5 min.

For immunofluorescence in MEFs, cells were cultured on 20-mm coverslips and fixed with 4% PFA. Blocking was done using 10% goat serum or donkey serum (Jackson Immuno) in PBS with 0.15% Triton X. Secondary antibodies were Alexa fluorophore-labeled goat/donkey IgG (1:250, Invitrogen). We used primary antibodies against: FLAG (1:500, Sigma F7425), Plk1 (1:2)[68], Plk1-phospho-T210 (1:200, Abcam ab39068), Pericentrin (1:3000, Abcam ab4448), γ-Tubulin (1:1000, Sigma T6557), Aurora B-phospho-T232 (1:500, Rockland 600–401–677), RacGAP1-phospho-Ser170 (1:200, Active Motif 39265-66), Sgo1 (1:100, S. Taylor, University of Manchester, UK), α-tubulin (1:500, Sigma T6199), and α-tubulin-FITC conjugate (1:500, Sigma F2168). Analysis of images was performed using Fiji. For the quantification of mean fluorescence intensity, all images were converted to 8-bit grayscale following which cell borders were traced using the free hand tool in Fiji and mean pixel intensity for corresponding channel was calculated within the defined area.

**Tumor cell culture**. Mammary tumors were digested with the Mouse Tumor Dissociation Kit (130-096-730, Miltenyi) in a gentleMACS Dissociator (130-095-937, Miltenyi) following manufacturer's instructions. Cells were cultured on 8-well chambered coverglass (Thermo Scientific, 155411) in DMEM supplemented with 10% tet-Free Serum, 4mM L-glutamine (GIBCO), 100 μg/ml Penicillin/Streptomycin (GIBCO), 5 μg/ml insulin (Sigma), 10 μg/ml EGF (Sigma), 1 μg/ml hydrocortisone (Sigma), 1 μM progesterone (Sigma), 5 μg/ml prolactin (NIHPP), and 1 μg/ml doxycycline. Time-lapse imaging during 15 h was performed on a Zeiss Cell Observer with 2 μm optical sectioning across 18 μm stack, 30 frames/h. Zeiss Zen 2 software served for image analysis.

**Human cell line culture**. MCF10A (a kind gift from Prof. Brummer) were maintained in DMEM/F12 (LONZA) supplemented with 5% bovine serum (Invitrogen), 1% Penicillin-Streptomycin (Life Technologies), 500 ng/ml hydrocortisone (Sigma-Aldrich), 100 ng/ml cholera toxin (Sigma-Aldrich), 10 μg/ml insulin (Sigma-Aldrich), and 20 ng/ml epidermal growth factor (EGF, Sigma-Aldrich). To generate PLK1 inducible cell line, MCF10A were first infected with an rtTA-expressing retrovirus and selected with puromycin (1 μg/ml), and then infected with an inducible Tet-ON lentivirus carrying the human Plk1cDNA cDNA (pLenti CMVtight Hygro DEST from Addgene #26433) and selected with hygromycin (350 μg/ml). Immunofluorescence and quantification of binucleation was performed as described with MEFs.

**RNA and protein work**. For RNA analysis, snap frozen tissue was homogenized using mortar and pestle while maintaining temperature at −80 °C using dry ice. RNA extraction was carried out following manufacturer's recommendations (RNeasy Mini Kit, Qiagen). QuantiTect Reverse Transcription Kit (Qiagen) was used for cDNA synthesis using 400 ng of RNA. Quantification using real-time PCR was initiated using 12 ng of cDNA with SYBR Green PCR Master Mix (2×) (Applied Biosystems) in a LightCycler II® 480 (Roche). The oligonucleotides for *Plk1* amplification were: F-(5′-3′) AACACGGCCTCATCCTCTACAAT and R: (5′-3′) AGGAGGGTGATCTTCTTCATCA. Primers used for monitoring Kras expression are listed in ref. [5].

For protein extraction and immunoblot, mouse tissues and MEFs were lysed in Laemmli lysis buffer (0.25 M Tris-HCl pH 6.8, 2.5% glycerol, 1% SDS, and 50 mM DTT). Samples were then boiled for 10 min and cleared by centrifugation. Proteins were separated on XT Criterion Tris-Glycine acrylamide gels (BioRad), transferred to nitrocellulose membranes (BioRad), and probed using the following specific antibodies: anti-FLAG (1:500, Sigma F7425); anti-Plk1 (1:500, Millipore Thermo Fisher 33–1700); anti-p53 (1:200, Cell Signaling 2524); anti-p21 (1:100, Santa Cruz sc-397); anti-phospho Ser10 H3 (1:1000, Cell Signaling 9701); anti-α-Tubulin (1:10,000, Sigma T9026); or anti-vinculin (1:10,000, Sigma V9131). Signal detection was done using secondary antibodies coupled to Alexa 680 dye (1:2000, Invitrogen) and using the Odyssey Infrared Imaging System (Li-Cor Biosciences). All original and full size western blots are shown in Supplementary Fig. 9.

**Three-dimensional organotypic assays**. Mammary glands were harvested from 8–9-week-old virgin female mice and three-dimensional organoid cultures were prepared according to published records[33]. Time-lapse imaging was performed during 20 h on an inverted spinning disk confocal (Perkin Elmer Ultraview-Vox): 0.3 μm optical sectioning across 35 μm stack, 5 frames/h and for 12 h using a Leica SP8 Confocal microscope with a resonant scanner: 0.9 μm optical sectioning across 35 μm stack, 12 frames/h.

**Human breast cancer data**. TCGA RNA-Seq expression data and clinical data were obtained from the TCGA data portal. Copy number and mutation data were obtained from ref. [69]. *PLK1* expression was normalized relative to *TBP* and log2 transformed. Genome doubling was estimated as previously described[70]. Survival analysis was performed using Kaplan–Meier curves and Cox proportional hazard models.

**Statistical analysis**. GraphPad Prism 6 was utilized for all statistical testing. Control samples for in vivo mammary gland-related experiments were obtained from animals containing the transgenes but kept on a normal diet or animals lacking MMTV-rtTA but placed under a doxycycline-enriched diet (625 ppm; Harland). Non-induced cultures (–Dox) were considered as controls in vitro.

**Data availability**. The data that support the findings of this study are available within this article and Supplementary Files, or available from the authors upon request.

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

## Acknowledgements

We are indebted to Stephen Taylor for the Sgo1 antibody. We thank Simone Kraut, Jessica Steiner, and the DKFZ light microscopy unit for excellent technical assistance. The results published here are in part based on data generated by TCGA pilot project (https://cancergenome.nih.gov/) established by the NCI and the National Human Genome Research Institute. The data were retrieved through dbGaP authorization (accession no. phs000178.v9.p8). S.V.V. was supported by the Marie Curie Network Ploidynet, funded by the European Union Seventh Framework Programme (FP7/2007–2013) under Grant Agreement #316964. L.S. is supported by a postdoctoral fellowship from Fundacion Ramon Areces. Work in the R.S. laboratory was supported by an ERC starting grant (#281614), Marie Curie PCIG09-GA-2011–293745 and the Howard Hughes Medical Institute. G.d.C. is funded by AECC Scientific Foundation (LABAE16017DECA). Work in the M.M. laboratory was supported by grants from the MINECO (SAF2015–69920-R cofunded by ERDF-EU), Worldwide Cancer Research (WCR no. 15–0278), and Comunidad de Madrid (iLUNG-CM; B2017/BMD3884). The CNIO is a Severo Ochoa Center of Excellence (MINECO award SEV-2015-0510).

## Author contributions

G.d.C., A.E.B., P.M., M.S., and B.E. generated the mouse model and analyzed the phenotype in MEFs. S.V.V., L.S., K.S., and K.R. generated the in vivo data and analyzed the phenotype in MEFs, mammary gland organoids, and tumor cells. A.d.M. performed the pathology analysis. N.McG. performed the analysis of human breast cancer. G.d.C., R.S., and M.M. designed and supervised the study and analyzed data. M.M. and R.S. wrote the manuscript with the help of G.d.C. and S.V.V.

## Additional information

**Competing interests:** The authors declare no competing interests.

