## [Peer Review File · Nature Communications]

Reviewers' comments:

Reviewer #1 (Remarks to the Author):

PIK1 is a multi-functional kinase with functions during and outside of mitosis. PIK1 gained significant attention due to the fact that it is frequently overexpressed in various cancer entities and consequently, PIK1 inhibitors were identified and are currently tested in clinical trials. Although elevated levels of PIK1 are often seen in cancer the functional consequences of its overexpression are insufficiently understood.

de Carcer et al. addressed this question by generating an inducible PIK1 overexpression mouse model (which is not the first model generated, see e.g.: Li et al., JBC 2017, a study which is not mentioned in this manuscript). Using transgenic MEFs they demonstrate that PIK1 overexpression results in various mitotic defects including premature chromatid separation, chromosome missegregation and cytokinesis failure. Based on the extensive work done already by many laboratories on the functions of PIK1 these observations are somewhat expected. Surprisingly, these mitotic defects are not associated with higher tumor incidence after PIK1 overexpression, although it is not clear whether this is due to insufficient PIK1 expression levels (see below).

In MEFs, the mitotic defects observed including polyploidization are associated with reduction of cell proliferation and –as the authors claim- also with a reduction of cell transformation induced by KRAS. Whether this is indeed caused by the mitotic defects remains unclear.

About 40% of the PIK1 overexpressing MEFs exhibit cytokinesis defects, which were addressed in more detail in this study. The authors show that failure of cytokinesis correlates with a reduced localization of Cep55, a known PIK1 substrate, and TSG101 (an ESCRT-I component) to the cleavage furrow. Again, whether this defect is causal for cytokinesis failure in these cells was not proven.

Finally, the authors show that overexpression of PIK1 reduces tumorigenesis and tumor growth in two tumor models (KRAS-G12D and HER2), which is meant to be the key finding of this study.

Overall, the authors address an important question regarding the consequences of PIK1 overexpression in tumors and they provide several pieces of interesting data. However, it remains completely unclear whether the different observations are causally related to each other. It is plausible that PIK1 overexpression causes mitotic defects. However, it is also clear that the same condition can induce additional non-mitotic defects as the same group has recently published (de Carcer et al., Nature Med 2017). Further, a recently published study demonstrated a role for PIK1 overexpression in the DNA damage response and this study showed that PIK1 overexpression can foster tumor development upon DNA damage, which somewhat contradicts the findings of this study (Li et al., JBC 2017). Hence, there are multiple possibilities how PIK1 overexpression can affect tumor development and there is no attempt to dissect these different possibilities in this study.

Rather, in this study, the authors focus on the induced defects in cytokinesis as the main cause leading to suppression of tumor growth. But this claim is not worked out. All phenotypes described (missegregation, bi-nucleation, reduced localization of the ESCRT complex etc) only correlate with reduced cell or tumor growth, but is it really causal? This key question remains unanswered. To support the claim, rescue experiments where a specific defect is selectively suppressed are required. For instance, if Cep55 loading is impaired and if this is really triggered by hyper-phosphorylation by PIK1 (is Cep55 really phosphorylated at higher level?) then this specific defect should be rescued by expression of phosphorylation mutants of Cep55. Without providing clear evidence for a causal relationship all observations are just correlations and could simply reflect bystander effects.

If polyploidization is indeed the main cause for tumor suppression in this context then I would expect that concomitant loss of p53, which clearly fosters polyploidy (Suppl. Figure 2e), would even more support tumor suppression. I doubt that this is the case.

It is also unclear whether overexpression induces apoptosis or whether the observed phenotypes of reduced tumor growth are due to proliferation inhibition.

Another problem of this study is the question of general importance. It is well known that mouse cells are much more prone to polyploidization in comparison to human cells. For example, loss of p53 causes polyploidization in mice, but not in human. Hence, it is questionable if Plk1-induced polyploidization is of any relevance for tumor development/suppression in humans. The authors should provide at least some experiments in non-transformed and e.g. KRAS-transformed human cells to provide evidence that similar outcomes of Plk1 overexpression (e.g. of suppression of transformation) are also relevant in human cells.

Finally, there are a number of technical issues and inconsistencies in this study, which limits my enthusiasm for this work. It is not clear why some of the experiments were done with the heterozygous, and some with the homozygous Plk1 mice. In Figure 1e, the authors used only the heterozygous mice to claim that Plk1 overexpression has no (or minor) impact on tumor-free survival. How about mice with homozygous status of Plk1 transgene? Are they tumor-prone? It is mentioned that only intermediate to HIGH levels of Plk1 (in MEFs) results in defective cell proliferation and polyploidy (Fig. 2a-d), but why was this condition (in homozygous mice) not investigated with respect to tumor development?

The heterozygous mice should exhibit aneuploidy, a condition which was shown to be often (not always) associated with tumorigenesis, especially in the lung. Did the authors specifically look at lung tumors in aged mice as well?

The status of the Plk1 transgene does not correlate well with the actual expression of Plk1 in different tissues (Suppl. Figure 1a). For example the heterozygous mice do not show Plk1 expression in the kidney while homozygous mice have no expression in the spleen. This is very inconsistent and raises the question whether Plk1 is robustly expressed at all in these mice. Does this contribute to the different phenotypes observed?

In Figure 3 it is shown that 40% of the cells exhibit polyploidy and/or binucleation. In contrast, in Figure 6, below 10% of the cells show tetraploidy. Why is there such a huge difference and how can this rather rare phenotype be responsible for such a dramatic tumor suppression as shown in Fig. 6c? Thus, it seems very unlikely that tetraploidy as a result of cytokinesis failure is the main mechanisms of tumor suppression.

Overall, it is interesting that co-expression of Plk1 in established tumor models results in suppression of tumorigenesis. However, the mechanisms behind this observation remain elusive and should be worked out before publication in Nature Comm. can be recommended. In addition, evidence should be provided supporting the notion that this tumor suppression is relevant in human tumors. If so, one should expect that tumors overexpressing Plk1 should generally have a better prognosis, for instance in lung or breast cancer where KRAS and HER2 hyper-activation are highly prevalent. Based on a large literature, this seems not to be the case.

Minor comments:

1. The labeling of figures 6d-g is mixed up in the text.
2. In the introduction the CIN70 signature is described as a gene signature predicting CIN. Meanwhile it is clear that this is not the case. This signature rather reflects proliferation (see work from A. Amon and others).
3. For Suppl. Figure 2a it is described that Plk1 is localized at kinetochores. This is not visible in the Figure.
4. It is unclear whether homo- or heterozygous Plk1 mice were used for the experiments in Fig. 6.

Reviewer #2 (Remarks to the Author):

This manuscript is focused on elucidating the role of Plk1 in regulating mitotic progression and tumorigenesis using the mouse genetic approach. Carcer et al have shown that Plk1 overexpression leads to abnormal chromosomal segregation and cytokinesis rather than enhanced cell proliferation. These authors also show that at the molecular level mitotic defects are mediated by defective loading of Cep55 and ESCRT complexes to the abscission bridge during cytokinesis. Significantly, enhanced levels of Plk1 greatly suppressed development of mammary glands tumors induced by either KRASG12D or Her2, which is accompanied by increased rates of chromosomal instability. Overall, the current study is of great significance as it demonstrates that Plk1 has a tumor suppressive property.

In general, mouse studies were well designed and executed. The dataset is clean with appropriate controls. Although mechanistic insights about how overexpression of Plk1 leads to cytokinesis failure are somewhat weak the current study does provide a solid biological role of Plk1 in tumor development. The following are several points to improve the work:

1. Figure 2a: Additional markers of mitosis should be included (e.g., p-H3S10, cyclin B)
2. An early study shows that Plk3/Prk shares similar property as Plk1 in rescuing CDC5 deficient yeast strain (J Biol Chem. 1996, 9;271:19402-8). Consistent with the current work, mammalian Plk3 also exhibits a tumor suppressive role in mice (Cancer Res. 2008;68:4077-85). These studies should be cited.
3. Figure 4d: Sgo1 staining of the cell with Dox treatment appears to be a bi-nucleated cell. There are differences in the intensity of staining among condensed chromosomes within the cell. Thus, better images would be helpful. What about cohesin levels (plus/minus Dox)? It is also helpful to show Plk1 staining in these cells.

Response to the reviewer's comments:

We would like to thank the reviewers for their constructive comments and suggestions as well as their positive feedback. We have performed additional experiments to address the reviewer's concerns, which have advanced the mechanistic details, quality and overall impact of our study, as well as changed the text and figures to provide a clearer interpretation of the results.

Below we include a point-by-point answer to their questions (in blue).

Point-by-point response to the reviewer's comments and suggestions

Reviewer #1 (Remarks to the Author):

Plk1 is a multi-functional kinase with functions during and outside of mitosis. Plk1 gained significant attention due to the fact that it is frequently overexpressed in various cancer entities and consequently, Plk1 inhibitors were identified and are currently tested in clinical trials. Although elevated levels of Plk1 are often seen in cancer the functional consequences of its overexpression are insufficiently understood.

de Carcer et al. addressed this question by generating an inducible Plk1 overexpression mouse model (which is not the first model generated, see e.g.: Li et al., JBC 2017, a study which is not mentioned in this manuscript). Using transgenic MEFs they demonstrate that Plk1 overexpression results in various mitotic defects including premature chromatid separation, chromosome missegregation and cytokinesis failure. Based on the extensive work done already by many laboratories on the functions of Plk1 these observations are somewhat expected.

The paper by Li et al., JBC 2017 was published after we had our manuscript sent to *Nature Communications*. We have included this new reference in the revised version of our manuscript. Although some of the observations regarding mitotic abnormalities are similar, please note that the main message in our paper, a tumor suppressor effect of Plk1 overexpression in cancer, is not addressed nor predicted from that manuscript, neither from others where Plk1 function as tumor suppressor has been evaluated before.

1. Surprisingly, these mitotic defects are not associated with higher tumor incidence after Plk1 overexpression, although it is not clear whether this is due to insufficient Plk1 expression levels (see below)

The reviewer raises a very interesting point. We do not think lack of tumors is due to insufficient levels of Plk1 due to two different observations.

First, we now include in the manuscript new data showing that higher levels of Plk1 expression, resulting from 2 copies of rtTA or 2 copies of Flag-Plk1, are detrimental *in vivo* and animals suffer from morphological aberrations in several tissues but, importantly, also in the absence of tumors. These new data has been included in the revised Supp. Figure 1b,c.

Second, some of the few tumors we obtained in the (+/Plk1);(+/rtTA) mouse cohort are actually negative for Plk1 transgene expression, indicating that tumors actually select lower doses rather than very high Plk1 levels (see an example below).

Liver (A) and lung (B) tumors are negative for Plk1 staining (as compared to induced expression in the neighbor healthy tissue), and more importantly are also negative for the Flag-tag staining, indicating that the Plk1 transgene is not expressed in the tumor tissue. We observed similar data in some of the histiocytic sarcomas and some lymphomas.

2. In MEFs, the mitotic defects observed including polyploidization are associated with reduction of cell proliferation and –as the authors claim- also with a reduction of cell transformation induced by KRAS. Whether this is indeed caused by the mitotic defects remains unclear

The reviewer is right and we are afraid that, in practical terms, there is no way to demonstrate a direct causal relationship. Due to the pleiotropic nature of Plk1 effects, we cannot rescue mitotic defects to analyze a putative recovery in proliferation. The only conclusion we take from the data is that Plk1 overexpression results in multiple mitotic abnormalities (Figures 3, 6, 7) and reduced proliferation in the presence of polyploid/aneuploid cells. We have revised the text to avoid any possible over-conclusion.

We have additionally performed new experiments that support a correlation between mitotic defects caused by Plk1 overexpression and reduced proliferation. For example:

- a) We have performed a new transformation assay in primary MEFs with HRas-V12 and EIA. After foci were formed, we evaluated the growth of these foci in the absence and presence of doxycycline. As shown in the revised Suppl. Figure 2g, Plk1 expressing foci grew much slower on doxycycline (with high Plk1 expression: see wb Suppl. Figure 2f) compared to transformed foci off dox. In addition, 25% of the Plk1 expressing cells continue to become binucleated (see revised Suppl. Figure 2h).
- b) We have also analyzed the fate of transformed MEFs with Ras-V12 oncogene in soft agar experiments. Concomitantly with the growth curve, transformed MEFs cannot generate colonies in soft agar in the presence of doxycycline. Moreover, we can also stop colony cell growth when the transformed colonies are already formed (revised Figure 2h). These demonstrate that Plk1 overexpression plays as a tumor suppressor even when cells are already tumorigenic.

Overall, although we agree with the observation made by the reviewer, we believe that this set of experiments strongly suggests a correlation between Plk1 overexpression and reduced proliferative/oncogenic potential. Please see additional discussions below regarding the pleiotropic effect of Plk1 in these cells.

3. About 40% of the Plk1 overexpressing MEFs exhibit cytokinesis defects, which were addressed in more detail in this study. The authors show that failure of cytokinesis correlates with a reduced

localization of Cep55, a known Plk1 substrate, and TSG101 (an ESCRT-I component) to the cleavage furrow. Again, whether this defect is causal for cytokinesis failure in these cells was not proven.

We thank the reviewer for this comment. As discussed above, Plk1 overexpression results in multiple abnormalities that prevent us from rescuing the general phenotype using a single downstream molecule. For this reason, we decided to rescue some of these phenotypes by inhibiting Plk1 transiently, and we demonstrate that the aberrant localization of Cep55 and Tsg101 actually depends on Plk1 overactivity (Figure 5c,d). We also tried to express the Cep55 Ser436 phospho-mutant form (see also below). Unfortunately, the mere expression of this construct results in abnormal cytokinesis as already reported (JCB, 2010. vol 191(4): 751-760). We have updated the text to indicate that we are not proposing Cep55 as the only Plk1 substrate involved in the effects described in the manuscript.

4. Finally, the authors show that overexpression of Plk1 reduces tumorigenesis and tumor growth in two tumor models (KRAS-G12D and HER2), which is meant to be the key finding of this study. Overall, the authors address an important question regarding the consequences of Plk1 overexpression in tumors and they provide several pieces of interesting data. However, it remains completely unclear whether the different observations are causally related to each other. It is plausible that Plk1 overexpression causes mitotic defects. However, it is also clear that the same condition can induce additional non-mitotic defects as the same group has recently published (de Carcer et al., Nature Med 2017). Further, a recently published study demonstrated a role for Plk1 overexpression in the DNA damage response and this study showed that Plk1 overexpression can foster tumor development upon DNA damage, which somewhat contradicts the findings of this study (Li et al., JBC 2017). Hence, there are multiple possibilities how Plk1 overexpression can affect tumor development and there is no attempt to dissect these different possibilities in this study.

The referee raises here interesting questions regarding the diverse Plk1 functions. We believe that the mitotic defects are indeed the mayor tumor suppressive mechanism since we robustly observe the same data in vitro and in vivo (i) increase in ploidy, (ii) increase in mitotic aberrations, (iii) increase in multinucleation. We agree that we cannot rule out other possible effects, as Plk1 is being related to cell migration and metastasis (Elife. 2016 Mar 22;5. pii: e10734. doi: 10.7554/eLife.10734.), regulates autophagy (Autophagy. 2017 Mar 4;13(3):486-505) and coordinates biosynthesis by directly activating the pentose phosphate pathway (Nat Commun. 2017 Nov 15;8(1):1506. doi: 10.1038/s41467-017-01647-5.). We feel that dissecting all the reported functions of Plk1 in these tumor models would be technically challenging and would require a huge amount of time. Regarding the specific effect that Plk1 deficiency has in the arteries (Nat Med. 2017 Aug;23(8):964-974. doi: 10.1038/nm.4364), we have not found any alteration in big arteries in these tumor models (data not shown), and small blood vessels in the tumors are likely not affected as they do not contain smooth muscle, which is the Plk1 target in its cardiovascular function. We have added a comment in this regard in the discussion.

Regarding the recent published paper in JCB (Li et al., JBC 2017), authors show here that there is something else Plk1 needs to be tumorigenic. In this case, as the reviewer highlights, DNA damage by gamma-irradiation is needed to foster cell transformation in the presence of Plk1. We do not claim Plk1 plays uniquely as a tumor suppressor, but that the mere expression of Plk1 is not sufficient to induce cell transformation and prevents proper tumor development, at least in the models included in our manuscript. Probably, if cells overexpressing Plk1 are challenged (as Li et al., have done), there is a mechanism by which Plk1 can synergize with a tumor driver. As this paper was published after

our original manuscript was submitted, we have now added additional discussion to cover these possibilities.

Rather, in this study, the authors focus on the induced defects in cytokinesis as the main cause leading to suppression of tumor growth. But this claim is not worked out. All phenotypes described (missegregation, bi-nucleation, reduced localization of the ESCRT complex etc) only correlate with reduced cell or tumor growth, but is it really causal? This key question remains unanswered. To support the claim, rescue experiments where a specific defect is selectively suppressed are required. For instance, if Cep55 loading is impaired and if this is really triggered by hyper-phosphorylation by Plk1 (is Cep55 really phosphorylated at higher level?) then this specific defect should be rescued by expression of phosphorylation mutants of Cep55. Without providing clear evidence for a causal relationship all observations are just correlations and could simply reflect bystander effects.

The reviewer is right in this recurrent point but, as discussed above, we are afraid that the pleiotropic nature of the alterations induced by Plk1 overexpression (as also reported in the new manuscript suggested by the reviewer) make these rescue effects highly unprovable. In addition, we do not claim that cytokinesis is the “main cause leading to suppression” as almost every mitotic abnormality may have antiproliferative effects. We have revised the text to make these points much clearer.

In any case, following the reviewer’s recommendation, we have tried to rescue the phenotype with the CEP55 phospho-mutant. However, expression of mutants in the Ser436 residue of CEP55 already rise cytokinesis problems since this mutant localize CEP55 too early in the cytokines furrow, therefore generating cytokinesis aberrancies. This issue is also commented in the original paper describing the Plk1 and CEP55 negative regulation [JCB, 2010. vol 191(4): 751-760] where authors address the following: “*Cep55 S436A was prematurely recruited to the central spindle and then accumulated to higher levels than the wild-type protein at the midbody (Fig. 3 B). Typically, Cep55 was recruited 60 min after the onset of anaphase, whereas Cep55 S436A was visible on the central spindle after 5–10 min (Fig. 3 B, bar graph). These cells remained arrested at the midbody stage for many hours*”.

This is the reason we made the rescue experiments using the Plk1 inhibitor for short periods of time at the end of the Dox induction. These rescue assays with the Plk1 inhibitor allowed us to partially rescue (due to the short time of inhibition) the binucleation, and more importantly CEP55 and TSG101 localization at the midbody and, more importantly, generating less binucleated cells, therefore rescuing the phenotype. Unfortunately, none of these experiments can be done in long term analysis (neither in vivo) as alteration in the levels of Cep55 is deleterious for the cells.

5. If polyploidization is indeed the main cause for tumor suppression in this context then I would expect that concomitant loss of p53, which clearly fosters polyploidy (Suppl. Figure 2e), would even more support tumor suppression. I doubt that this is the case.

This is an interesting point. Unfortunately, these assays crossing additional alleles until tumors appear would require more than 1-2 years to be completed.

Following the reviewer’s suggestion, we decided to look at human breast cancers with known p53 status. As shown in the revised Figure 8b, PLK1 expression is significantly higher in polyploid tumors with mutant p53 compared to non-genome doubled tumors, suggesting mutations in p53 may allow for higher PLK1 expression in tumors with ploidy alterations.

6. It is also unclear whether overexpression induces apoptosis or whether the observed phenotypes of reduced tumor growth are due to proliferation inhibition.

The reviewer raised an important question here. We never observed any significant increase in cell death in MEFs overexpressing Plk1. On the other hand, we observed increase in senescent cells due to high levels of polyploidy (Suppl. Fig. 2e). Nonetheless, we have tested in a new experiment the apoptosis marker Annexin V in MEFs treated with Dox for 1 and 3 days. We do not observe any significant increase in apoptosis upon Plk1 overexpression. These new data are now incorporated in the revised Suppl. Figure 2c.

In vivo, in healthy mammary glands, we observe a slight increase in apoptosis when Plk1 is overexpressed during 4 days (Old Suppl. Figure 5e, now revised Suppl. Figure 7e).

Following the reviewer's comment, we have also looked for cell death in the mammary tumors in both oncogenic models (Kras and Her2). We stained these tumors with caspase 3 as a marker of apoptosis and we see no clear differences between the tumors induced by the oncogene alone or in combination with Plk1 overexpression. We then stained the same tumors with PCNA as a marker of proliferation. These data showed a significant reduction in the number of PCNA positive cells in the tumors arising from Her2/Plk1 and Kras/Plk1 compared to Her2 or Kras alone. To further confirm the reduced proliferation in these tumors, we stained them with a p21 antibody and here again we confirmed a significant increase in p21 levels in tumors from Her2/Plk1 and Kras/Plk1 compared to Her2 or Kras alone. These data are now provided in the revised Suppl. Figure 6 c,d.

7. Another problem of this study is the question of general importance. It is well known that mouse cells are much more prone to polyploidization in comparison to human cells. For example, loss of p53 causes polyploidization in mice, but not in human. Hence, it is questionable if Plk1-induced polyploidization is of any relevance for tumor development/suppression in humans. The authors should provide at least some experiments in non-transformed and e.g. KRAS-transformed human cells to provide evidence that similar outcomes of Plk1 overexpression (e.g. of suppression of transformation) are also relevant in human cells.

We agree with the reviewer that this is an important point. In light of his/her suggestion, we have used the non-transformed breast cell line MCF10A expressing rtTA and infected them with an inducible Plk1 vector. Results show that overexpression of Plk1 in these cells leads to an increase in binucleation similar to mouse cells. This new data has been added to Suppl. Figure 2i. Further, we also used the transformed breast cancer cell line MDA-MB-231 with a known mutation in KRas-G13D expressing an rtTA. Here after infection with an inducible Tet-ON lentivirus carrying the human Plk1 cDNA, the % of binucleated cells increased from 8% (-Dox) to 20% (+Dox).

In addition, in accordance with the reviewer's helpful suggestion, we investigated the importance of PLK1 expression in human breast cancer datasets. First, we investigated whether Plk1 expression was significantly different between genome doubled (GD) and non-genome doubled (nGD) tumors. As shown in the revised figure 8a, PLK1 levels were significantly higher in GD tumors compared to nGD ones. This data suggests that is associated with polyploidization in human breast cancer. Importantly, this was found to remain significant in the context of wildtype TP53. We have added these important findings to the revised manuscript.

We then explored the clinical relevance of PLK1 expression. Consistent with a tumor suppressive role for PLK1, we found that tumors with low PLK1 expression were associated with significantly reduced overall survival. Importantly, this remained significant in multivariate analysis, including PAM50 subtypes, genome doubling and TP53 status. This data is now included in the revised Figure 8.

8. Finally, there are a number of technical issues and inconsistencies in this study, which limits my enthusiasm for this work. It is not clear why some of the experiments were done with the heterozygous, and some with the homozygous Plk1 mice. In Figure 1e, the authors used only the heterozygous mice to claim that Plk1 overexpression has no (or minor) impact on tumor-free survival. How about mice with homozygous status of Plk1 transgene? Are they tumor-prone? It is mentioned that only intermediate to HIGH levels of Plk1 (in MEFs) results in defective cell proliferation and polyploidy (Fig. 2a-d), but why was this condition (in homozygous mice) not investigated with respect to tumor development?

We apologize for the confusion we might have created due to the lack of sufficient details in the initial version of the manuscript. In addition to heterozygous mice, we have now completed the analysis of homozygous animals on doxycycline for tumor development. Interestingly, these animals die early at a median latency of 20 days after doxycycline administration, due to severe alterations in the intestinal and colon tissue architecture, leading to an impairment of nutrient uptake and a subsequent dramatic loss of weight. These mice also show severe aplasia in most of proliferative tissues such as spleen, bone marrow, skin, etc. Reduction in proliferation is also reflected in the peripheral blood cell population analysis where there is a reduction in red blood cells (RBC), white blood cells (WBC) and lymphocytes (LYM). Of note, no increased tumor susceptibility is observed in these homozygous mutants. These data is now included in the revised Suppl. Figure 1b,c,d.

9. The heterozygous mice should exhibit aneuploidy, a condition which was shown to be often (not always) associated with tumorigenesis, especially in the lung. Did the authors specifically look at lung tumors in aged mice as well?

We only found one lung tumor in our ageing assays, but this lung tumor is negative for the Plk1 transgene expression as we show above in point number 1. We also observed bronchiolar epithelia dysplasia in the Plk1 overexpressing animals as depicted in Figure 1f of the original manuscript.

10. The status of the Plk1 transgene does not correlate well with the actual expression of Plk1 in different tissues (Suppl. Figure 1a). For example the heterozygous mice do not show Plk1 expression in the kidney while homozygous mice have no expression in the spleen. This is very inconsistent and raises the question whether Plk1 is robustly expressed at all in these mice. Does this contribute to the different phenotypes observed?

The reviewer is right about misleading information in Suppl. Figure 1a. Hence, we have repeated this expression analysis with three additional animals per cohort, and with consistent genotypes similar to those used in the study. Additionally, to Plk1, we have included the Flag-tag immunodetection.

As reflected in the paper where this technology is described (Beard et al. *Genesis* 44, 23-28 (2006)), expression of the transgene varies according to different tissues, and there are certain tissues more prone to express the transgene than others. We also observe this in our Plk1 transgenic mice. As the new figure shows, kidney is one of the tissues where we see lower levels of expression, and when analyzed by histology Plk1 is restricted to glomerulus (see figure 1d). This is the reason we barely observe expression by WB. Similarly happens in spleen, where we observe very little Plk1 expression. On the other hand, there are some other tissues where Plk1 is abundantly overexpressed such as pancreas, intestine, thymus, lung and liver.

Noteworthy, when we analyze the expression of Plk1 in (+/Plk1);(+/-rtTA) mice, we observe variations in expression levels in between different animals, a finding common in many transgenic models. Accordingly, to the IHC staining, we observe patches of expression into the same tissue (as reported in Beard et al., 2006) and this might explain this variability. In order to clarify this point, we have now added, in figure 1d panel, an immunostaining for Flag on several (+/Plk1);(+/-rtTA) mouse tissues.

11. In Figure 3 it is shown that 40% of the cells exhibit polyploidy and/or binucleation. In contrast, in Figure 6, below 10% of the cells show tetraploidy. Why is there such a huge difference and how can this rather rare phenotype be responsible for such a dramatic tumor suppression as shown in Fig. 6c? Thus, it seems very unlikely that tetraploidy as a result of cytokinesis failure is the main mechanisms of tumor suppression.

We apologize that this analysis has led to misinterpretations. In the previous version of Figure 6g, we counted as tetraploid cells only those where we saw 4 or more dots. We thank the reviewer for realizing that indeed this is not correct, as tetraploidy can be an intermediate state of triploidy and therefore in the tumors we may have many triploid cells that are not counted as tetraploid. We have now repeated the analysis and calculated the % of cells with 2 dots, 3 dots or 4 dots and more. We clearly see that Her2/Plk1 have 30% of cells with 3, 4 or more dots, compared to 13% in Her2 alone tumors while Kras/Plk1 have 18% compared to 3% in Kras. See revised Figure 6f.

It must be noted also, that here the reviewer is comparing MEFs to tumor tissue, adding to the fact that time lapse microscopy is far more reliable to detect polyploidy occurrence compared to FISH. The point that we are making here is that tumors with high Plk1 levels have more tetraploid cells compared to oncogene alone tumors.

In addition, as an independent method to address polyploidy, we have revised the nuclear volume measurements of tumor cells. Tumor sections were visualized under a TissueFAXS slide scanning platform while the quantitation was performed using StrataQuest software (TissueGnostics). The automated measurements of the nuclear area allowed us to count a minimum of 8.000 cells per tumor sample, again verifying that Kras/Plk1 tumors as well as Her2/Plk1 tumors have a significantly increase in nuclear volume compare to the single oncogene tumors. This data has been incorporated in the revised Suppl. Figure 6b.

Finally, we also performed a new experiment in which tumor cells were isolated from Her2 or Her2/Plk1 tumors. After a short period in 2D (overnight), tumor cells were followed by time-lapse microscopy for a median of 14 hours to measure mitotic defects. As shown in the revised Figure 6, 24% of Her2 tumor cells display mitotic errors compared to 50% in Her2/Plk1 tumor cells. While the

main mitotic defects observed in Her2 tumor cells were lagging and misaligned chromosomes (19%), in Her2/Plk1 we observed 22% of cells failing to complete cytokinesis and 3% with anaphase failure.

Again, we do not have data to claim that cytokinesis failure is the main defect leading to tumor suppression and we have revised the text to clarify that combination of multiple mitotic defects contribute to this phenotype.

12. Overall, it is interesting that co-expression of Plk1 in established tumor models results in suppression of tumorigenesis. However, the mechanisms behind this observation remain elusive and should be worked out before publication in Nature Comm. can be recommended. In addition, evidence should be provided supporting the notion that this tumor suppression is relevant in human tumors. If so, one should expect that tumors overexpressing Plk1 should generally have a better prognosis, for instance in lung or breast cancer where KRAS and HER2 hyper-activation are highly prevalent. Based on a large literature, this seems not to be the case.

As already mentioned in point 7, we have now included in the manuscript an analysis of human breast cancer datasets. In agreement with the reviewer's hypothesis, we observe that tumors with higher Plk1 expression generally have improved prognosis, compared to tumors with low levels of Plk1 expression. This data is now included in the revised version of the manuscript and in Figure 8

13. Minor comments:

1. The labeling of figures 6d-g is mixed up in the text.

We thank the reviewer for realizing this mistake. The labeling of the figure has been corrected.

2. In the introduction the CIN70 signature is described as a gene signature predicting CIN. Meanwhile it is clear that this is not the case. This signature rather reflects proliferation (see work from A. Amon and others).

We agree on this reviewer comment and therefore we have removed this paragraph.

3. For Suppl. Figure 2a it is described that Plk1 is localized at kinetochores. This is not visible in the Figure.

The figure panels in Suppl. Figure 2a have been changed to add better quality images to show the proper localization of Plk1 by means of IF against Flag-Plk1, as well as a new immunofluorescence with Plk1 antibody (revised Suppl. Figure 2b). Kinetochores can be seen now as decorating dots close to the chromosomes in the middle panel of Suppl. Figure 2a.

4. *It is unclear whether homo- or heterozygous Plk1 mice were used for the experiments in Fig.*

We are sorry for this confusion. All the experiments done in the mammary glands of transgenic animals and 3D culture systems of primary mammary cells were performed with heterozygous mice for all the transgenes (Plk1, Her2, Kras, MMTV-rtTA), except for H2B-GFP. We have added a more detailed description in the materials and methods section.

Reviewer #2 (Remarks to the Author):

This manuscript is focused on elucidating the role of Plk1 in regulating mitotic progression and tumorigenesis using the mouse genetic approach. Carcer et al have shown that Plk1 overexpression leads to abnormal chromosomal segregation and cytokinesis rather than enhanced cell proliferation. These authors also show that at the molecular level mitotic defects are mediated by defective loading of Cep55 and ESCRT complexes to the abscission bridge during cytokinesis. Significantly, enhanced levels of Plk1 greatly suppressed development of mammary glands tumors induced by either KRASG12D or Her2, which is accompanied by increased rates of chromosomal instability. Overall, the current study is of great significance as it demonstrates that Plk1 has a tumor suppressive property.

In general, mouse studies were well designed and executed. The dataset is clean with appropriate controls. Although mechanistic insights about how overexpression of Plk1 leads to cytokinesis failure are somewhat weak the current study does provide a solid biological role of Plk1 in tumor development. The following are several points to improve the work:

1. Figure 2a: Additional markers of mitosis should be included (e.g., p-H3S10, cyclin B)

As suggested by this reviewer we have now repeated the wb of Figure 2a to include phospho-histone H3 S10 (pH3), as a mitotic marker. There is no increase in pH3 when cells are treated with Dox, showing that cells overexpressing Plk1 do not arrest in mitosis for long periods of time and they are able to cycle.

2. An early study shows that Plk3/Prk shares similar property as Plk1 in rescuing CDC5 deficient yeast strain (J Biol Chem. 1996, 9;271:19402-8). Consistent with the current work, mammalian Plk3 also exhibits a tumor suppressive role in mice (Cancer Res. 2008;68:4077-85). These studies should be cited.

This is a very convenient suggestion by the reviewer and we have added and discussed these references in the revised version. In the same trend, we also have discussed that Plk5 also might play a role as a tumor suppressor.

3. Figure 4d: Sgo1 staining of the cell with Dox treatment appears to be a bi-nucleated cell. There are differences in the intensity of staining among condensed chromosomes within the cell. Thus, better images would be helpful.

We have now included a better picture of a diploid cell, showing the reduced levels of Sgo1 in Plk1 expressing MEFs. The new data is in the revised Figure 4d.

4. What about cohesin levels (plus/minus Dox)? It is also helpful to show Plk1 staining in these cells.

At the time of submission, we were unable to provide a working staining for cohesin because of technical difficulties and the lack of a working antibody for cohesin in mouse cells. Following the referee's comment, we tested cohesin loading in MEFs upon Dox addition, by chromatin fractionation. Firstly, Flag-Plk1 transgene is located at the chromatin fraction of MEFs upon Dox addition. When we test the loading of cohesin (Rad21) in cells treated with Dox during 72 hours, we were unable to observe any significant change in Rad21 loading at the chromatin fraction. Since we found no differences in cohesin loading, and due to space constrains, we decided to add here the data for the reviewer to see, but to not include this data into the revised version of the manuscript.

Chromatin fractionation

In addition, we have included a staining with a Plk1 antibody in primary MEFs (in metaphase and anaphase) in the absence and in the presence of doxycycline. The staining clearly shows the proper localization of Plk1 in cells as well as increased levels of Plk1 in Dox treated cells. These new data is included in the revised Suppl. Figure 2b.

We hope that our revisions properly addressed all the reviewers concerns and that our revised manuscript will now be found suitable for publication in Nature Communications.

With best regards,

Rocio Sotillo

Reviewers' comments:

Reviewer #1 (Remarks to the Author):

de Carcer et al. provide a revised version of their manuscript on the role of PLK1 overexpression in tumor progression.

In their work, the authors claim that PLK1 overexpression has a tumor suppressor function and that this is mediated by multiple mitotic defects. In the first version of their manuscript they focused more on the role of cytokinesis failure as a mediator of tumor suppression. Nevertheless, the work questions a large body of published data showing that PLK1 is overexpressed in many human cancers including breast cancers and demonstrating an oncogenic function of PLK1.

Although adding a few new data the work still suffers significantly from the lack of drawing clear conclusions. The authors try to relate abnormal mitosis as a consequence of PLK1 overexpression to a tumor suppressor function of PLK1. Unfortunately, this is still not supported by the data presented.

I can follow the point that PLK1 overexpression causes multiple mitotic defects. This includes the generation of lagging chromosomes indicating chromosome missegregation and cytokinesis failure leading to binucleated cells. These observations are somewhat expected since PLK1 is a key kinase in these processes and elevating the activity of a key mitotic kinase is expected to cause multiple defects. In addition, the authors show that also other defects, e.g. chromatin bridges are induced after PLK1 overexpression. The reason for that is less clear, but might hint to additional roles during DNA replication or in the DNA damage response pathway (e.g. implicated by the work of Li et al., 2017). Indeed, the authors show that PLK1 activity is increased in interphase cells after its overexpression. The role of DNA damage is not further addressed. Thus, the data suggest that PLK1 overexpression results in multiple and little defined mitotic (and possibly interphase) defects.

The authors additionally address the cytokinesis failure after PLK1 overexpression and provide little evidence for Cep55 being a target for PLK1. This is known for a long time and the authors do not add any new information here. No rescue experiments are provided, which would clearly show that Cep55 is the relevant target for this phenotype in the overexpression setting. According to the rebuttal letter this could not be addressed due to technical reasons. Other strategies were obviously not tested. Nevertheless, due to the (expected) cytokinesis failure polyploid cells are generated.

Altogether, these (not really new) data show that PLK1 overexpression causes more or less undefined mitotic defects.

The major problem with this work comes from the fact that the authors try to relate these ill-defined defects to a tumor suppressor function of PLK1. They show that (very strong) PLK1 overexpression causes reduced cell proliferation and reduced tumor growth in two breast cancer models. How this is related to mitotic dysfunction remains completely unclear. Again, no rescue experiments, not even in vitro are provided (e.g. rescuing cytokinesis defects).

In addition, there are severe inconsistencies:

(i) in the Kras model the mitotic defects are very little. Less than 20% of the cells show aneuploidy or polyploidy (Fig. 6e,f) while this model exhibit a profound (90%) increase in tumor-free survival of the mice (Fig 6a). How can this be explained?

(ii) in the Her2 model the aneuploidy/polyploidy is much higher (up to 30-40%, Fig. 6e,f), but the effect in terms of tumor suppression and survival is only 50% of that in the Kras model. This is not easy to recapitulate.

In principle, it is an interesting finding that PLK1 overexpression can suppress Kras or Her2 induced tumors. However, the mechanisms causing this tumor suppression are still not worked out. It remains unclear whether the mitotic defects are indeed responsible for this tumor suppression. And if so, is the aneuploidy or the polyploidy responsible for the observed effects? There are many examples where similar mitotic defects (and aneuploidy/polyploidy) support tumor growth. Thus, it may be obvious that PLK1 has additional functions (during interphase?) that contribute to its tumor suppressing function. This is not ruled out. Unfortunately, it also remains unaddressed whether the role of PLK1 is breast-specific or also seen for other cancer entities.

Recommendation: Without providing a link between mitotic defects and tumor suppression the work presented here remains a side-by-side story. Therefore, publication in Nature Com. cannot be recommended. A more cancer-specific journal would be more appropriate.

Additional comments:

1. The authors claim that they present a "new mouse model for PLK1 overexpression...", first heading of the results section. This is not true. Li et al., 2017 published already a PLK1 overexpression model. This paper is still hardly mentioned.
2. The authors still refer to the (wrong) CIN signature in the introduction
3. Fig. 1c shows localization of PLK1 on the spindle, which is NOT the "proper" localization (Results: first page, bottom line)
4. The lack of PLK1 overexpression in many tissues is puzzling and prevents the detection of tumorigenesis in these tissues. This is highly relevant e.g. for lung cancer where PLK1 overexpression is highly prevalent in humans. However, these mice do not express PLK1 in lungs.

Reviewer #2 (Remarks to the Author):

No additional comments